# Deep water hydrodynamic observations of two moorings sites on the continental slope of the Southern Adriatic Sea (Mediterranean Sea)

Francesco Paladini de Mendoza[1], Katrin Schroeder[1], Leonardo Langone[2], Jacopo Chiggiato[1], Mireno Borghini[3], Patrizia Giordano[2], Giulio Verazzo[2], Stefano Miserocchi[2]

[1]CNR-ISMAR, Arsenale Castello, 2737/F, 30122 Venezia (VE), Italy.
[2]CNR-ISP, Via P. Gobetti, 101 - 40129 Bologna (BO), Italy.
[3]CNR-ISMAR, Pozzuolo di Lerici, 19032 La Spezia, (SP), Italy.

*Correspondence to*: Francesco Paladini de Mendoza (francesco.mendoza@ve.ismar.cnr.it)

**Abstract.**

This work presents an 8-years long dataset of monitoring activities conducted on the western margin of the Southern Adriatic Sea, where two moorings have been placed since 2012 in sites that are representative of different morpho-dynamic conditions of the continental slope (open slope vs. submarine canyon). The dataset includes measurements conducted with both current meters and CTD probes, and provides information about the hydrodynamics and thermohaline properties of the last 100 m of the water column. The hydrodynamics in both sites is dominated by weak currents ($<0.1$ ms$^{-1}$), which undergo yearly to episodic pulsation able to exceed intensity greater than 0.5 ms$^{-1}$ which are linked to the passage of dense waters. The 8-years records presented here, represents a starting point for the continuous observation activity set up on occasion of the "Operation Dense Water" in 2012, focused on the Southern Adriatic deep-water dynamics. Since then, the observatory has been on-going since 2012 and the database is regularly updated. All the data described here are made publicly available from https://doi.org/10.5281/zenodo.6770201 (Paladini de Mendoza et al., 2022) and are compliant with the FAIR principles (findable, accessible, interoperable and reusable).

## 1 Introduction

The Adriatic Sea is a sub-basin of the eastern Mediterranean Sea, with unique characteristics from the geological and oceanographic points of view. The basin is enclosed between two mountain chains (The Apennines and Dinaric Alps) to the west and to the east and with an elongated shape (the axis is oriented from SE to NW), a length of 800 km and an average width of 180 km. In the south, the Adriatic Sea is connected with the Ionian Sea through the Strait of Otranto.

The Southern Adriatic Margin was built during the last half million years and its structure is a result of eustatic depositional cycles and complex quaternary uplift and deformation patterns (Bertotti et al., 1999; Ridente and Trincardi, 2002) well summarized by Bonaldo et al. (2016). The north-western sector of the slope is constituted by the south Gargano system which is the main deformation zone that extends offshore with the Gondola Deformation Belt (GDB, see Fig.1). In this sector there is a complex bedforms system where large sediment drift and furrow field give evidence of a strong bottom currents activity

(Minisini et al., 2006; Martorelli et al., 2010) as well as asymmetric upstream migrating mud-waves indicate a depositional environment (Verdicchio and Trincardi, 2006; Trincardi et al., 2007a, b; Verdicchio et al., 2007). To the south, the major conduit influencing off-shelf fluxes and deep circulation is provided by the Bari Canyon System engraved in the continental

margin with two main branches with a total length of about 30 km along the W-E direction (Turchetto et al., 2007; Rubino et al., 2012). During the sea-level low stand of the last glacial maximum, sediments were supplied into the canyon directly from river deltas or longshore drift, but at present the canyon head is far from the coastal sediments and can be fed only by shelf currents or episodic density currents.

The Adriatic Sea is one of the three Mediterranean sites where density currents can be originated and are essentially of two

typesthe North Adriatic Dense Water (NAdDW) a cold and dense shelf water that forms in the northern sector during intense and cold outbreaks in winter (Hendershott and Malanotte-Rizzoli, 1976; Franco et al., 1982) and the Adriatic Deep Water (ADW) which forms by open ocean convection, between late winter and early spring, in the center of the permanent cyclonic gyre of the Southern Adriatic and vertically mixes the water column up to a variable depth (Vilibić and Orlić, 2002; Gačić et al., 2002; Manca et al., 2002; Civitarese et al., 2005).

In February 2012 the European region experienced a two weeks severe cold period that heavily impacted the northern Adriatic Sea. Immediately a rapid response experiment, called Operation Dense Water, was conducted in the southern Adriatic to observe the dense water masses dynamics (Chiggiato et al., 2016b). These activities funded by the Italian research program RITMARE (https://maritime-spatial-planning.ec.europa.eu/projects/la-ricerca-italiana-il-mare) spanned from oceanographic modeling, physical and biogeochemical oceanographic observations to sedimentological analyses of the erosional and

depositional bedforms. In this context moorings were placed along the southern Adriatic basin and the location of the mooring sites was chosen on the basis of the most prospective passage of dense shelf water obtained through an integrated approach between modeling-based predictions and geology-driven inferences. From 2012 the monitoring was continued, leading to the collection of 8-years datasets of two sites placed in two different areas along the continental slope of the southern Adriatic Sea. The moorings are equipped with an ADCP-RDI system and CTD probes which measure currents along the last 100 m of the

water column and thermohaline properties. The moorings sites are in the western sector of the continental margin respectively at 700 m and 600 m depth in an open furrow area of the continental slopes and in the main channel of the Bari Canyon System. The two different morphologies of the mooring site make the datasets representative of two different dynamic conditions of the continental slope of the southern
basin.

This paper presents data collected from 2012 to 2020 of which short segments of the dataset relative to 2012 have been used in previous studies (Mihanovic et al., 2013; Chiggiato et al., 2016; Langone et al., 2016; Marini et al, 2016; Foglini et al., 2015; Carniel et al., 2016; Bonaldo et al., 2016; Cantoni et al, 2016) for the analysis of physical processes (Chiggiato et al., 2016) and particle dynamics (Langone et al., 2016) induced by the 2012 cascading events. The data collected from the moorings are part of the IFON network (Italian Fixed-Point Observatory Network, Ravaioli et al., 2016) and the Southern

Adriatic Sea observatory of the EMSO-Italy Joint Reasearch Unit (https://www.emsoitalia.it/south-adriatic-sea), and the

continuous measurement activity provides a unique observatory on the hydrodynamic processes along the Southern Adriatic slope that have a direct implication on water renewal and transfers of organic and inorganic particulate matter from the shelf to the deep sea.

## 2. Setting, instruments, data and methods

The data come from two moorings located along the continental slope of the Southern Adriatic Sea (Figure 1a). The mooring sites are placed in two different locations that differ from a geomorphological point of view. The mooring site called BB is placed at 600 m depth on the main branch of Bari Canyon System (BCS) at 41°20.456'N, 17°11.639'E while the mooring site FF is placed at 41°48.396'N, 17°02.217'E on open slope furrow area of the continental slope at 700 m depth. The stand-alone moorings are equipped with an ADCP system which measures currents along the last 100 meters of the water column and a

CTD probe located approximatively 10 m above the bottom. Moorings, still operative, were configured and maintained for continuous long-term monitoring following the approach of the CIESM Hydrochanges Program (www.ciesm.org/marine/programs/hydrochanges.html; Schroeder et al., 2013). The 110 m long mooring scheme is represented in Fig. 1b. The ADCP system measures the intensity and directions of currents along the water column and has a temperature sensor in its transducer head. The CTD probe provides measurements of temperature and salinity (along with pressure). The

measurements are extended from 2012 until 2020 and divided in separated deployments interspersed approximately every 6 months for instrumentation recovery, data downloading and maintenance.

The ADCP used is of the type RDI Workhorse (Teledyne RD Instruments USA, Poway, California), using a four-beam, convex configuration with a beam angle of 20° and a working frequency of 307 kHz. The instrument is moored at a mean nominal depth of 500 m (BB) and 600 m (FF) in downward-looking mode at roughly 100 m from the seabed. The number of depth

cells is set to 27 with a cell size of 4 m. The sampling interval is set to 1800 s with 45 ping per ensemble. An ADCP computes sound speed based on an assumed salinity and transducer depth and on the temperature measured at the transducer. The system measures water temperature at the depth of the transducer by means of a thermistor embedded in the transducer head between the four beams. The sensor provides measurements in a range between -5 - 45 °C, with a precision of ± 0.4 °C and resolution of 0.01 °C.

Approximately at 10 meters above the seabed there is a CTD probe, SBE 16plus V2 SeaCAT to record thermohaline parameters. The SBE 16plus V2 SeaCAT is a high-accuracy CTD recorder designed for moorings or other long-duration, fixed-site deployments. In addition, the probe is equipped with optional pump for bio-fouling protection. Data of water conductivity was measured by sensor, with accuracy of 0.0005 S/m and resolution of 0.00005 S/m; the water temperature by means of a thermometer, with accuracy of 0.005 °C and resolution of 0.0001 °C; the water pressure by means a pressure strain

gauge sensor with an accuracy of 0.002% of full-scale range.

The resulting dataset covers the period from 8 March 2012 to 26 June 2020 for both moorings (details about surveys are reported in Table 1 where there is the description about the temporal extensions of each measuring period, the mooring location, depth and S/N of the ADCP system used. The instrumentation recovery consists on data downloading, battery replacing and

sensor checking together with maintenance operations. Every mooring component (ropes, chains and hooks) are checked to
ensure the functionality, durability and resistance during surveying period The probes are cleaned from fouling and sensor state is checked. In Table 1 is reported the S/N of instrumentation used in each deployment useful to know when some probes have been replaced. The probes were calibrated at Sea-Bird Europe - European Calibration and Repair Center and in Table 2 is reported the date of calibration of each CTD probe. During the cruise of October 2018 and 2019 (detailed in the cruise report of Cardin et al., 2018 and Bubbi et al., 2019) a comparison profiles are carried out with a calibrated onboard CTD probe, in
order to ensure the measurement agreement. The onboard CTD is a SeaBird SBE911plus equipped with dual sensors of temperature, conductivity. The pressure, temperature and conductivity sensors were calibrated at Sea-Bird Europe in September 2016. The CTD was attached to a SBE32 Carousel Water Sampler with 24 bottles (10 lt) together with an altimeter. From three depth levels, depending on the vertical profile of the stations, water samples were taken, also for calibration purposes of the salinity values and they were analysed in laboratory using a Guildline Autosal Salinometer.
The distances between the deployment points of the moorings during the different time period did not exceed 140 m for BB and 306 m for FF.

The ADCP time series is not fully continuous not for instrument failures but for operational reasons linked to recovering and sailing during maintenance surveys. Interruptions occurred twice a year approximately every six months (at the end of winter and in autumn), trying to make them as short as possible. The CTD time-series follows the deployment of ADCP but sampling
strategy is not always in agreement with ADCP time-series as well as the continuity of the dataset.

The sampling interval in the BB site was always at 1800 s and continuity of the data reflected those of ADCP records except for a data gap, due to battery discharge, that extended from 31 July 2015 to 09 November 2015. In the FF site the sampling interval in the first survey was 600 s, in the 5[th], 9[th], 10[th] and 11[th] surveys the sampling interval was 3600 and in the 7[th] survey the sampling interval was 10800 s. The continuity of the data reflects those of ADCP records except for two data gaps, due to
malfunctioning and battery discharge, that extended from 14 August 2013 to 08 November 2013 and from 30 May 2014 to 02 November 2014.

## 2.2 Dataset and Metadata description

The dataset is composed by 4 files in NetCDF format containing observational data and related metadata from the two mooring sites, (BB and FF) for the period March 2012 - June 2020. Each mooring site has two datasets for ADCP and CTD data,
respectively, and each file name specifies the mooring site name, its depth and the instrument (ADCP or CTD).

The data and metadata information includes Global Attributes, Dimensions and Variables. Global Attributes contain a summary description of the dataset and details about their geospatial position, temporal extension and data interval, the institution responsible of measurements, principal investigator name and contact, the observational network to which the mooring belongs, the conventions ("OceanSITES v1.4,SeaDataNet_1.0,COARDS,CF-1.6")and keywords vocabulary
('SeaDataNet parameter discovery vocabulary") used. Dimensions have the available structure of variables. For each variable

is reported the size, the dimensions, the datatype and attributes which last give specific information about each variable such as the measured parameter, the unit, the sensor, the data value range etc.

Regarding the ADCP dataset it is provided details about the station name, the probe serial numbers, the geographical position, the time, the depth, the cell depth and the component east, north and vertical of the current speed. The dataset contains the original ADCP variables followed by the qualification flags according to the results of the quality control (QC) procedure described in the next chapter.. Regarding CTD are reported the original data and qualifiers flag which define the quality of temperature and salinity data. The headers of quality flag assigned for each variable are followed by the suffix "-_qc".

It is ensured that all data described here are findable, accessible, interoperable and reusable (according to the FAIR principles, Tanhua et al., 2019), since they are identified by a unique persistent identifier (see abstract and Data Availability section), which also allows them to be retrievable, with all metadata records being accessible as well. Permanent DOI and Metadata ensure Findability of the dataset. The permanent Accessibility of the dataset is allowed by the use of Zenodo public repository. The data are available in NetCDF format which ensure the Interoperability with other platforms. The data and metadata specified in the global attributes use the SeaDataNet parameter discovery vocabulary (https://www.seadatanet.org/Standards/Common-Vocabularies) and the conventions: OceanSITES v1.4,SeaDataNet_1.0,COARDS,CF-155 1.6..The metadata accurately describes the data ensuring their reusability in future research and their integration with other data sources.

## 2.3 Data quality check

A first visual check of ADCP and CTD data time series gives a quick idea about whether the data looks reasonable or not judging by the average values of the parameter measured and the overall 'noisiness' of the plot. This screening phase allows to detect anomalous values which are those out of range with the rest of the series and helps to exclude from the time series data when systems are outside the water determining the corrected start and end of the time series. One parameter that helps this first screening phase is the depth measured by the pressure sensor, which indicates when the probe is at the operating depth and when it is being recovered at the surface. Applying these checks, a maximum of 1.95% of data of BB and 1.48% of data of FF were removed from the dataset.

Regarding ADCP data a second step consists of the determination of the seabed and the portion of the water column with good data. The seabed is detected by a specified filter algorithm named as "side lobe interference" which is based on the principle that the echo through the side lobe facing the surface or the bottom returns to the ADCP at the same time as the echo from the main lobe at certain distance to the surface that depends on the beam angle. In the case of a beam with an angle of 20° this means data from the last 6% of the range to the bottom can be contaminated. When looking down, as in our case, the contamination from bottom echoes usually biases velocity data toward zero. The next data processing applies a data QC criteria based on the parameter "percentage good" , provided from the recording system and indicates the fraction of data passed a variety of criteria which include low correlation, large error velocity and fish detection (false target threshold). To ensure the robustness of the collected data we have used a threshold of 80% to define good data (PG80). The structure of the data matrix

explained in the published database metadata consists of the original data, to ensure data accessibility and reusability, and a

quality flag is assigned to each observation following the SeaDataNet QC guidelines (SeaDataNet, 2010) and the L20 SeaDataNet Measurand Qualifier flags (last updated at http://seadatanet.maris2.nl/v_bodc_vocab_v2/browse.asp?order=conceptid&formname=search&screen=0&lib=l20) as shown in Table 3.

As shown in Figure 2 where an example of quality control procedure is depicted, when the hydrodynamic data exceed the

PG80 threshold, the quality flag assigned corresponds to code 3 which indicate the value recognised as inconsistent after QC while when the data pass the QC, the assigned flag is 1. Flag 9 is assigned when data are missing in the original time series.

For CTD measurements, the raw hexadecimal were converted to ASCII by the SBE Data Processing™ and after visual inspection, QC tests were applied to the data according to SeaDataNet guidelines (SeaDataNet, 2010), which rely on a Spike (ST) and a Gradient Test (GT):

In the ST is evaluated if the differences between sequential measurements are too large (ST>6°C for temperature, ST>0.9 for salinity)

$$ST = | \ V2 \ \text{-} \ (V3 + V1)/2 \ | \ \text{-} \ | \ (V3 \ \text{-} \ V1) \ / \ 2 \ |$$

where V2 is the measurement being tested as a spike, and V1 and V3 are the previous and next value

In the GT is evaluated if the gradient between adjacent salinity and temperature measurements are too steep (GT<9°C for

temperature, GT>1.5 for salinity):

$$GT = | \ V2 \ \text{-} \ (V3 + V1)/2 \ |$$

where V2 is the measurement being tested, and V1 and V3 are the previous and next values.

The threshold considered in the Spike Test of the SeaDataNet procedure is a fixed value and extremely large when considering the variability of our data. Therefore, the quality control of temperature and salinity was implemented by defining an

appropriate threshold for the data based on statistical analysis. The threshold was obtained from the calculation of the IQR parameter (Hald, 1952), which is a parameter used in the identification of outliers in non-symmetrically distributed data. The IQR parameter is estimated from the $25^{th}$ (Q1) and $75^{th}$ (Q3) percentiles through the relationship IQR = Q3 - Q1, from which the threshold for identifying an outlier is obtained, which is equal to 3xIQR (3IQR). For each mooring, both the distributions of temperature and salinity data with the values corresponding to Q1 and Q3 are shown (Fig.3a), as well as graphs of the ST

values calculated for each dataset with the 3IQR quality control threshold reference (Fig.3b).

When QC computation is completed the time-series is organized with every observation followed by specific flag code according to the SeaDataNet qualifier flag: data that pass QC test are flagged with code 1, data that do not pass the test are flagged with code 4.

Applying these guidelines, no anomalies and spikes were found in the dataset. In the dataset all data are reported except when

the probe is outside the water and not at the correct mooring depth. The flag 9 is assigned when data is missing.

## 3. Results

### 3.1 Thermohaline records

Temperature is measured by ADCP and CTD at two different depths along the water column respectively at roughly 100 m above the bottom (mab) and 10 mab. The time series shown in Fig. 4 in both sites starts in March 2012 after the cold air
outbreak occurred in the northern Adriatic and the time series starts during cascading events well known in literature (Chiggiato et al., 2016b).

In the upper layer (100 mab) the temperature recorded by ADCP in the BB site has a mean value of 14.02+/-0.27°C with a minimum temperature of 12.93°C and maximum of 14.95°C. In the lower layer (10 mab) the CTD highlighted an average temperature of 13.92+/-0.24 with a minimum temperature of 12.57°C and maximum of 14.78°C.

In FF at 100 mab the mean recorded temperature is 13.89+/-0.19°C with a minimum temperature of 13.12°C and maximum of 14.49°C while near the bottom (10 mab) the measurements indicate average value of 13.64+/-0.26°C with a minimum temperature of 12.10°C and maximum of 14.17°C.

Observing the total time-series of Fig. 4 the two sites have synchronous fluctuations more marked in the BB site. In addition, in the BB site the temperature slight differences can be appreciated between the two measurements depth while in the FF site
the temperature closest to the bottom is generally lower and has more pronounced variations. The time-series show a periodicity of water-cooling with an almost constant annual frequency but variable between years. The most marked events besides 2012 are 2013, 2017 and 2018.

Regarding salinity measurements in the BB site in the lower layer the average salinity is about 38.81+/-0.04, (minimum of about 38.59 and maximum 38.97). In the FF site, the salinity records at 10 mab have a mean value of 38.78+/-0.05 PSU
(minimum of about 38.64 PSU and maximum of 38.95 PSU).

The total time-series of salinity in Fig.5 has more marked variations in the BB site where the salinity is generally higher than the FF site. From 2018 a positive trend is appreciable in both sites and less differences between sites occur.

The temperature data in Fig. 6a are represented to give a quick overview of the inter- and intra-annual variability of the data. In the scatter plot the temperature is distributed along the x-axis and separates different months by colors. In the upper layer
of the FF site, the variations are restricted in a narrow range while in the lower layer wide temperature fluctuations are visible and always concentrated between February and June. In BB the vertical variability is less marked but the time-window when temperature decreases coincides. Statistics about temperature records grouped by months and years are reported in Tables 3 and 4 analyzing mean and maximum differences between upper (ADCP) and lower (CTD) layers. Generally, the temperature differences between ADCP and CTD layer is constrained around 0.05°C and 0.2°C in both sites and experience from January
to May largest decrease of water temperature especially close to the seabed. While in BB the maximum temperature difference between upper and lower layer is 0.73°C, in FF this can exceeds 1°C. The most intense cooling of water occurs in both sites during 2012, 2013, 2017 and 2018 when vertical difference increase. During 2019 only in the BB site cool events occur with

less intensity than the others. On annual scale large vertical temperature differences occur in FF while in BB the difference is less evident.

Salinity data are represented in Fig. 7 in the same way as temperature. In this case observations are limited only to the layer close to the seabed where the CTD probe is moored. In the FF sites the variations are restricted in a narrow range except during 2012 but in both sites the salinity decrease is always concentrated between February and June. Statistics about salinity records grouped by months and years are reported in Table 5. Generally, the variation of mean salinity between months is very narrow (<0.02) in both sites but between February and June salinity has got the maximum decrease of more than 0.1. On an annual

scale the largest variations occur in both sites during 2012, 2013, 2017 and 2018. In the BB site high variability of salinity is also observed in 2015 and 2016.

### 3.2 Hydrodynamic records

In this section we present the hydrodynamic measurements along the water section measured by the ADCP from 2012 to 2020 in BB site and in the FF site. In order to detail the dynamic variability along the water column the water column is separated

in the three vertical layers (roughly to 1/3 of the measured water column): Upper Layer (UL), Intermediate Layer (IL) and Lower Layer (LL). In the polar histogram the directions are binned every 5° and speed is divided in three classes.

*BB site*

Figure 8 shows the 8-years-long ADCP records at the BB site as vertical distribution of the speed module along the 23 layers

of the water column (Fig. 8a), as polar histograms (Fig. 8b) and as polar scatterplots (Fig. 8c) which represent the direction and intensity of currents along the water column. Generally, the current field is very weak ($0.07+/-0.01$ ms$^{-1}$) but during episodic energetic events the flow may exceed 0.5 ms$^{-1}$. The polar histograms represent hydrodynamic climate (Fig. 8b) where speed and directional class are clustered to represent the occurrence probability of the events while in Fig. 8c it is possible to observe the magnitude and direction of every single event scattered on a polar diagram. The hydrodynamic field of the three

layers highlights currents which spread between 100 - 225°N with a reigning directional sector in the upper layer between 170 and 200°N. The directional spreading of currents assumes a clear bimodal behavior approaching toward the seabed with reigning currents SSW oriented and dominant currents oriented toward SE. This behavior is more marked at the bottom and indicates a flow oriented toward the canyon axis to 110° or southward along the direction of the isobath (Chiggiato et al., 2016). This is a robust feature of this location (Turchetto et al., 2007) where currents directed along canyon are directly

associated to cascading flow while southward flow is indirectly associated to cascading as geostrophically adjusted downslope flow (Chiggiato et al., 2016). The diagrams b and c of the LL explain clearly the dynamic of along canyon axis currents which dominate in terms of intensity despite their low contribution in terms of frequency.

Figure 9 shows the time series of the currents of the UL and LL represented with a"moving average" 7-daily smoothing which allows a better visualization of the data by highlighting the vertical variability of currents especially during periods of flow

intensification.The average speed of the currents is $0.069+/-0.005$ ms$^{-1}$ in the UL and $0.079+/-0.006$ in the LL and the flow

accelerations when they occur involve the entire observed water column reaching a maximum speed of 0.76 ms⁻¹ in the LL and 0.58 ms⁻¹ in the UL. The  maximum difference between UL and LL reaches a maximum value of 0.38ms⁻¹.

An acceleration of the flow occurs approximately every year between February and May. The intensification of the current field varies year by year and reaches the greatest magnitude in 2012 and 2018 on the contrary of the weakest during 2014 and 2015. During energetic events a general increase of the current speed toward the seabed is visible and the components of flows have positive values for the east component and negative for the northern. The vertical component has small values in a range of -0.01 – 0.05 ms⁻¹ and is mainly directed toward the bottom during current pulses. The behavior of components reflects the direction of flow appreciable in the polar plots (Fig. 8c) which are directed mainly toward S and SE.

*FF Site*

In the site FF, the 8-years records show an average weak hydrodynamic field with value of 0.05+/-0.01 ms⁻¹ able to reach speed until 0.79 ms⁻¹ during the episodic strong current pulses (Fig. 10a). As observed in the BB site, the pulses of currents in FF recur in a temporal window every year (between February and May). The three layers represented in the polar histogram plots of Fig. 10b (constructed in the same way of BB) details the vertical variability of the flow along the water column. The flow in the UL is southward within a directional range centered to 180°N with more than 99% of the datasets below the intensity of 0.2 ms⁻¹. Proceeding down in the IL the directional spreading of currents becomes narrow (always centered to 180°N) and the intensity slightly increases, remaining always below 0.4 ms⁻¹. In the LL intense currents are clearly visible (magnitude greater than 0.6 ms⁻¹) directed toward south-east (150°N) in addition to the contour-parallel background current regime directed southwards. These intense events, with a very scarce frequency, indicates ageostrophic dynamics determined by the steepness of the continental margin which allows to break the geostrophic constraints (Chiggiato et al., 2016) flowing downward on open slopes responsible of the origin of furrow marks reported in this site by Trincardi et al. (2007a). In the time-series of the upper and lower layers, the currents speed (Fig. 11) has an average value very similar 0.049+/-0.003 ms⁻¹ (UL) and 0.057+/-0.012 ms⁻¹ (LL) but during pulses the speed increase concentrate at the bottom. During these events the velocity difference between LL and UL can reach 0.51 ms⁻¹.

During the acceleration phases of currents, the components of flow have a positive increment of eastern component together with a greater negative acceleration of the northern component. The eastern component in the UL never increases sensibly, while near the seabed it has the greatest increment. The vertical component is very weak (<0.05 ms⁻¹) but during the flow acceleration in the bottom layer its positive values suggest a flow directed toward the seabed.

## 4. Conclusions

The data presented here are the results of 8-years monitoring activities conducted on the western margin of the Southern Adriatic Sea where two moorings have been placed since 2012 in two sites of the continental slope representative of two different morpho-dynamic conditions of the Southern Adriatic Margin influenced by the passage of dense shelf water.

Long-term high-resolution monitoring in sensitive area such as zones of dense water passage and deep waters constitute a key element in the monitoring network of the current context of global climate change, improve the oceans understanding and shed light on its complexity. The measurement site of this dataset represents a stable node of the European observatory system EMSO-ERIC consortium. The measurement site is one of the EMSO regional facility for the South Adriatic Sea located in the western part of the basin with the objective to assess the Adriatic's response to climate forcing.

The moorings, equipped with ADCP and CTD probes, provide measures of hydrodynamic and thermohaline parameters on a section of the water column extended for the last 100 m from the seabed.

In occasion of the extreme severe cold outbreak in north Adriatic occurred in 2012 was set up the "Operation Dense Water" which have produced wide literature about the dynamic of cascading events (great part grouped in the special issue edited by Chiggiato et al., 2016b) and their linked processes. The observatory has been continued until today with the aim to answer the several questions unaddressed. Some open questions are related to the frequency of cascading events and their magnitude variability in a long-time scale. This data block extended from 2012 to 2020, to represent a starting point for broadening the knowledge and thus giving even more robustness to previous research results about the Southern Adriatic deep-water dynamic. Generally speaking, the 8-years' time-series are characterized in both sites by reigning weak currents ($<0.1$ ms$^{-1}$) which undergo yearly to episodic pulsation able to exceed intensity greater than $0.5$ ms$^{-1}$. These pulsations are linked to the passage of dense waters with low temperature and salinity which exhibit in both sites an intra- and inter-annual variability. During the year, the oceanographic effects of the passage of these currents are extended over a six months window where the core is concentrated between February and May. These dense water masses that originated several months earlier (Vilibilic and Orlic, 2002; Vilibic and Supic, 2005; Chiggiato et al., 2016) can flow along the slope in the southern sector until June with a progressive weakening of the intensity. Due to the distance from the generation area the Adriatic dense water propagation, unlike other sites of dense water generation (i.e Gulf of Lion), requires more time to reach the southern slope where cascading may occur and the start of passage of dense water flow depends on the onset of the generation. For example, in 2012 first pulses of dense water were observed as early as three weeks after its generation in February (Benetazzo et al., 2014).

In the FF site the flow has a clear dominant direction (140-150°N) especially in the bottom layer. Along the profile the currents undergo sharp intensification and rotate toward the main direction which takes a definite direction only in the lower layers. The dynamic observed in FF is already described as a peculiar site behavior where the dense water flow is organized in multiple short-lived pulses with short duration (Chiggiato et al., 2016). This dynamic leave traces in the morphology of the FF site where extensive presence of abyssal furrow, documented by Verdicchio et al., 2007, are indicator of strong and directionally currents (Bonaldo et al., 2016), are oriented (145°N) according to the direction of the currents. The time-series can contribute to answer still open questions about deep water dynamic and in particular to processes related to dense water passage. The continuous monitoring is fundamental to improve the knowledge about the dense water formation processes, water mass properties, biogeochemical cycles, and cascading in the southern Adriatic, understanding ecosystem function especially in relation to carbon sequestration dynamics and acidification processes in deep waters and in future releases we plan to include comparison with other existing data and climatology. The collected parameters are of great interest for different disciplines,

ranging from geosciences to physical oceanography, to biogeochemistry and marine ecology and recorded variables can be used to compare different sites and other project can benefit from this monitoring site for the configuration of new monitoring stations.

## 5. Data availability

All data is made publicly available through the Zenodo repository. The registered database DOi is https://doi.org/10.5281/zenodo.6770201 (Paladini Mendoza, et al., 2022).

This paper describes in detail the temporal coverage of the dataset which is constituted by quite continuous high temporal resolution time series of currents, temperature and salinity from 2012 to 2020. The adopted methodology about mooring configuration and data records, quality control procedures ensure compliance and consistency of the dataset and represent the largest deep-water observatory of current and thermohaline data of the Southern Adriatic Sea. The dataset presented conclude in 2020 but monitoring activities are still in progress and future data collected by these stations will be added to an updated version of the repository as advancing of the data collection to convey the progress of oceanographic observations to the scientific community. The strategy is to have an update of the dataset every two years.

## 6. Acknowledgements

This paper is realised in the context of PRIN-PASS project. The maintenance of BB and FF fixed moorings over time was only possible thanks to the support of various projects: European Community's Seventh Framework Programme projects HERMIONE (Grant agreement No. 226354) and COCONET (Grant agreement No.287844) of the European Commission, the Flagship project RITMARE SP5_WP3_AZ1 (the Italian Research for the Sea). This work was supported also by the EMSO-Italia Joint Research Unit (JRU). The authors thank the cruise participants who helped us with the mooring servicing, in particular the captain and the crew members of the R/V's Urania, Minerva Uno, G. Dallaporta, Laura Bassi and OGS Explora, and of the fishing boats Pasquale & Cristina, and Attila.

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

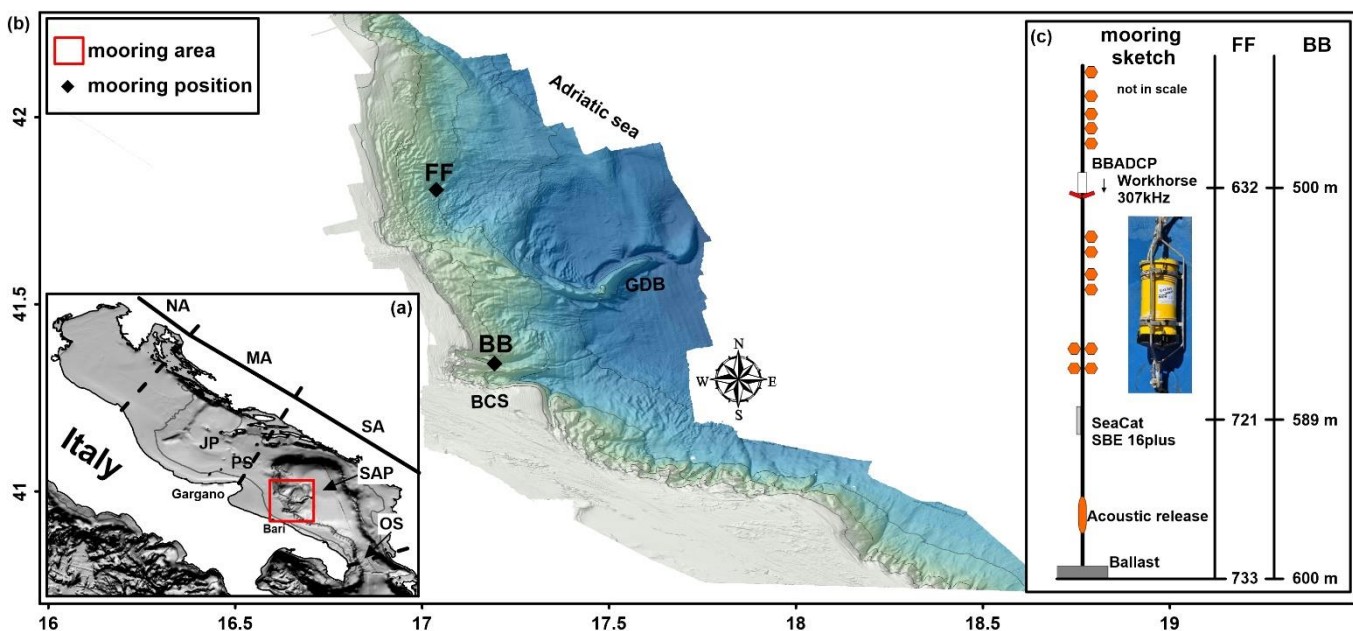

Figure 1: Study Area. panel (a) represents the Adriatic Sea divided by dotted lines in the three sub sectors North Adriatic (NA),
Middle Adriatic (MA) and South Adriatic. JP indicates Jabuka Pit, PS the Pelagosa Sill, SAP is South Adriatic Pit and OS is the
Otranto Strait. The red box encloses the western margin where moorings are deployed detailed in panel; bathymetry is provided by
EMODNET portal (https://portal.emodnet-bathymetry.eu/) (b) where BB and FF are respectively the mooring site in the Bari
Canyon System (BCS) and in the Open Slope. The GDB is the Gondola Deformation Belt. The high-resolution bathymetry is obtained
from EMODNET portal. The panel (c) represents the sketch not in scale of mooring structure.

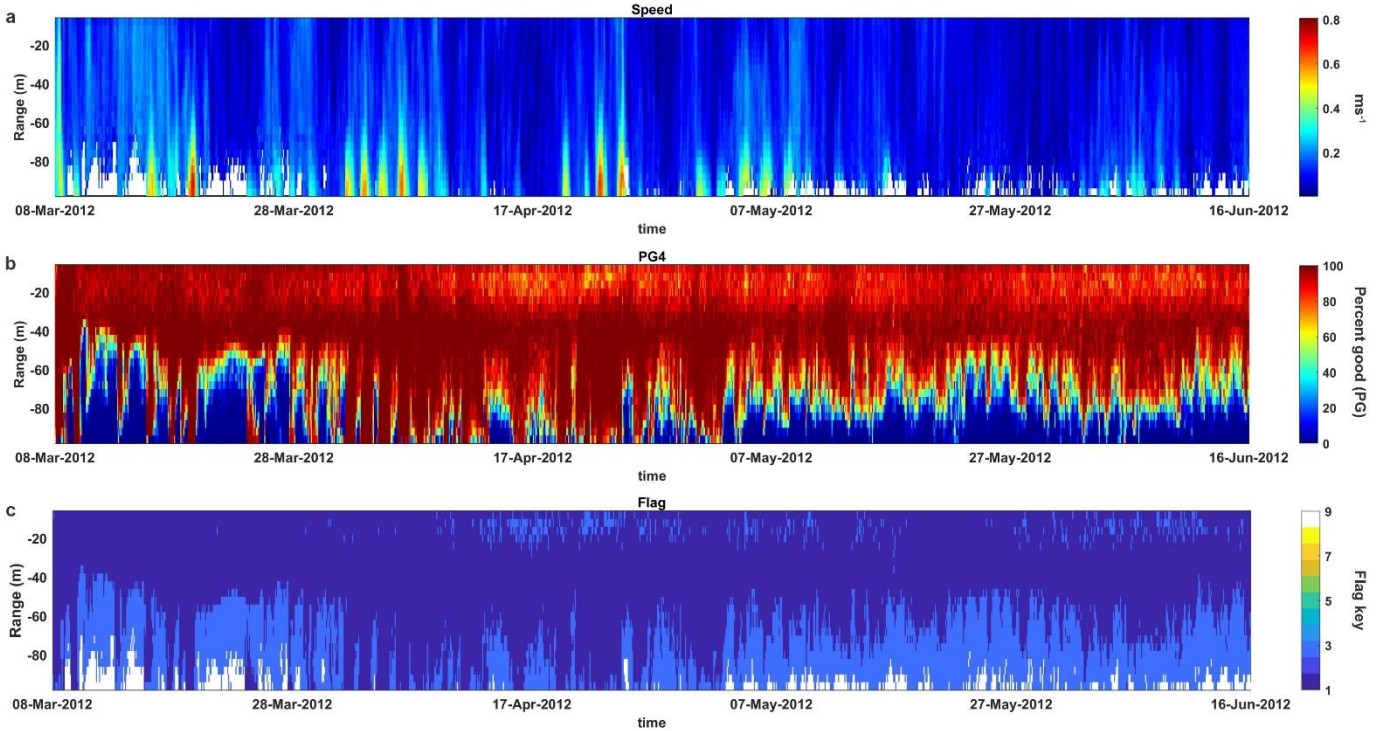

**Figure 2: Schematic of the quality control procedure: in** *a* **are represented the original current velocity data, in** *b* **the value of the Percent Good parameter calculated for each cell, and in** *c* **the key of the qualifier flag assigned for each observation. For better visualization, only part of the time series has been represented.**

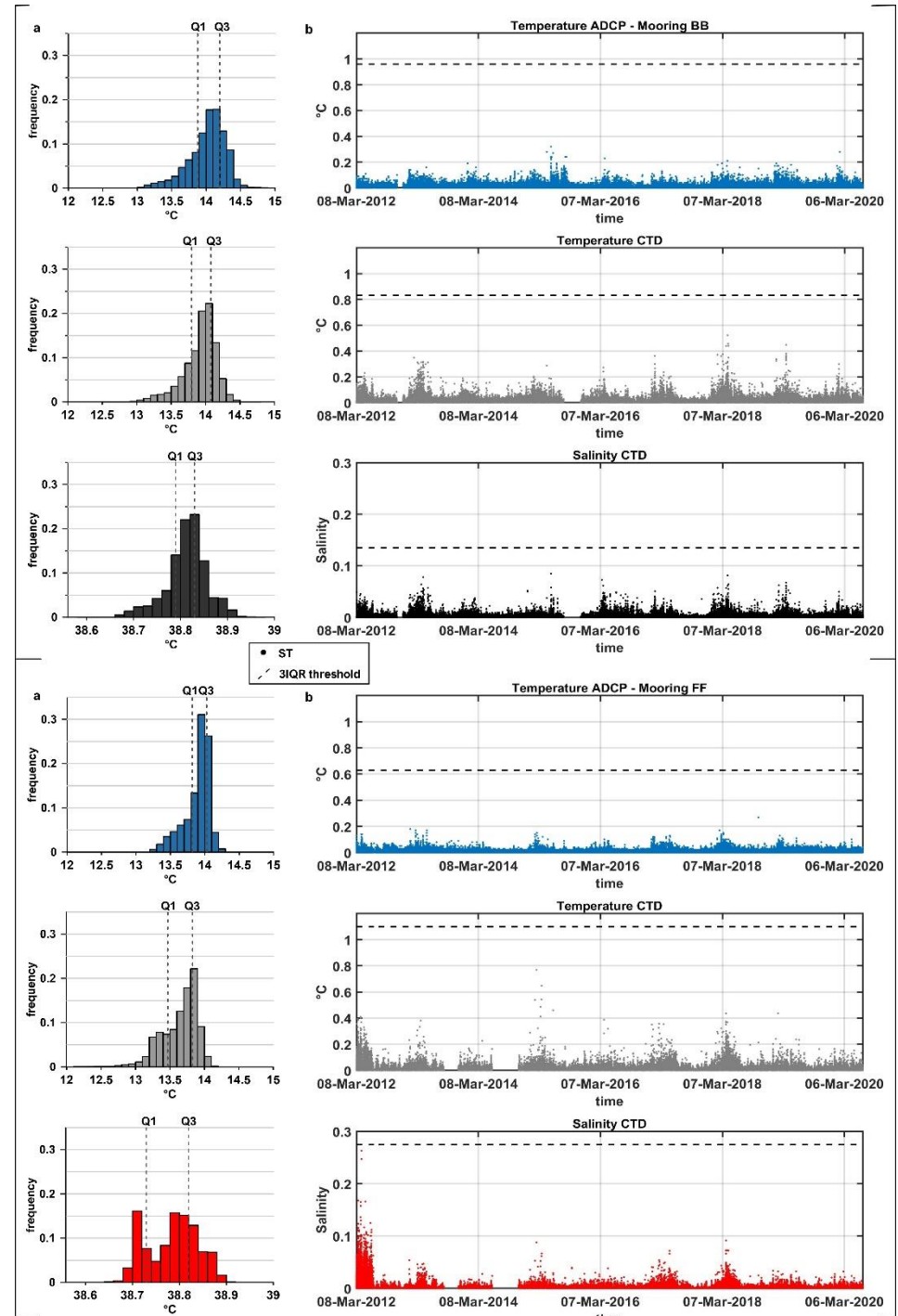


**Figure 3: Quality control of temperature and salinity data measured by ADCP and CTD. a) the distributions of the temperature and salinity data recorded at each berth are shown, and Q1 and Q3 represent the 25th and 75th percentiles used for IQR calculation;**

**b) the time series of the ST value calculated for each observation (dots) and the 3IQR value (dashed line) indicating the threshold for identifying outliers are shown.**

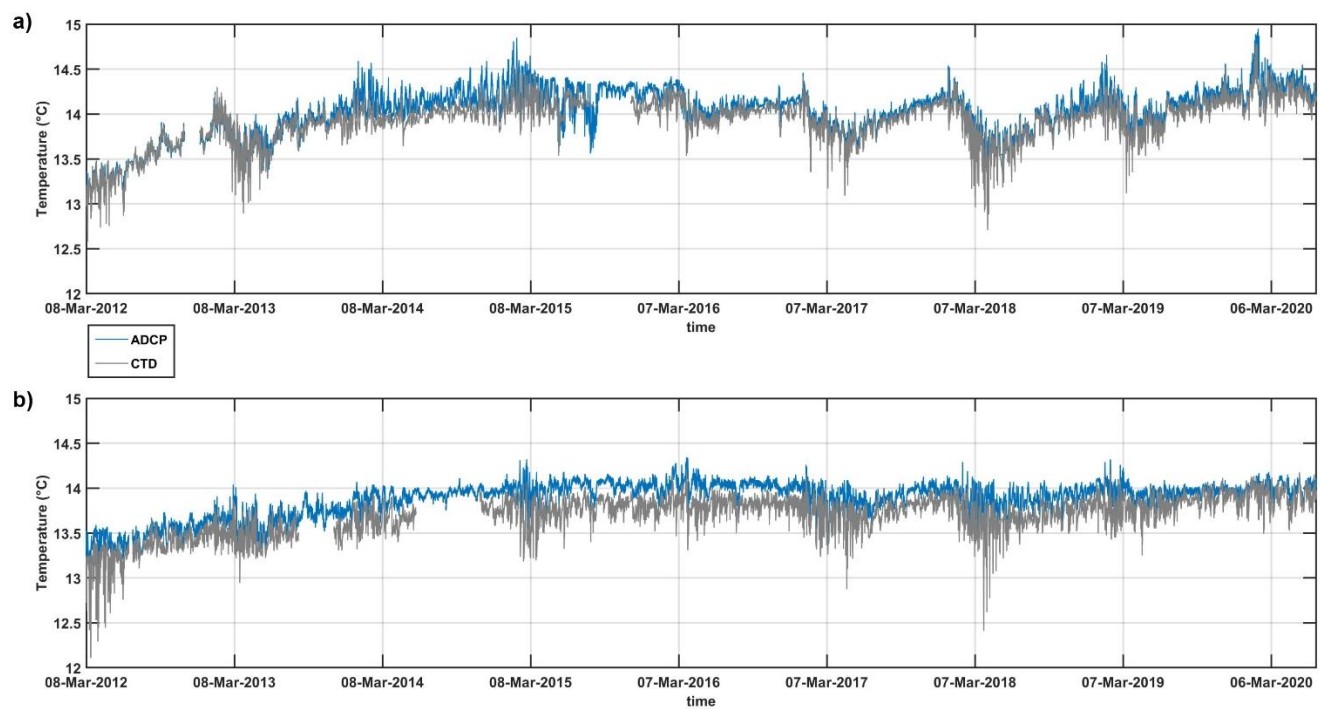


**Figure 4: ADCP and CTD temperature records at two mooring sites (a) BB on canyon (600 m depth), (b) FF on the open slope (700m).**

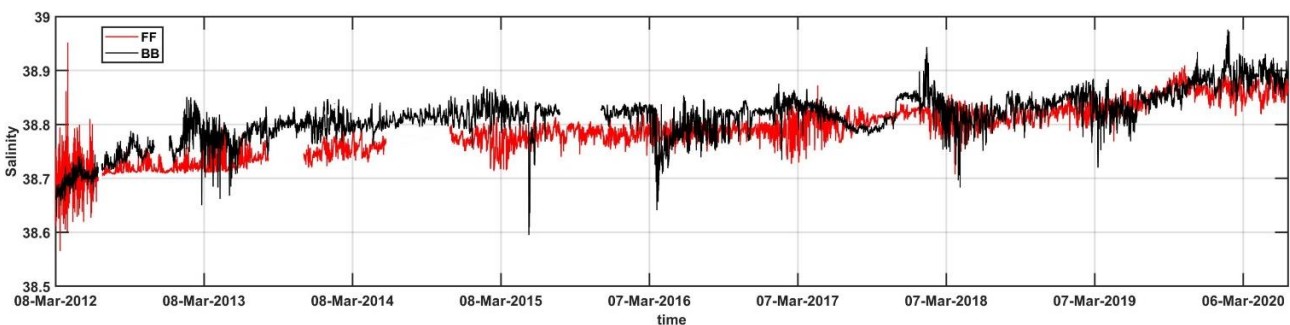

**Figure 5: CTD Salinity records on the two mooring sites BB and FF..**

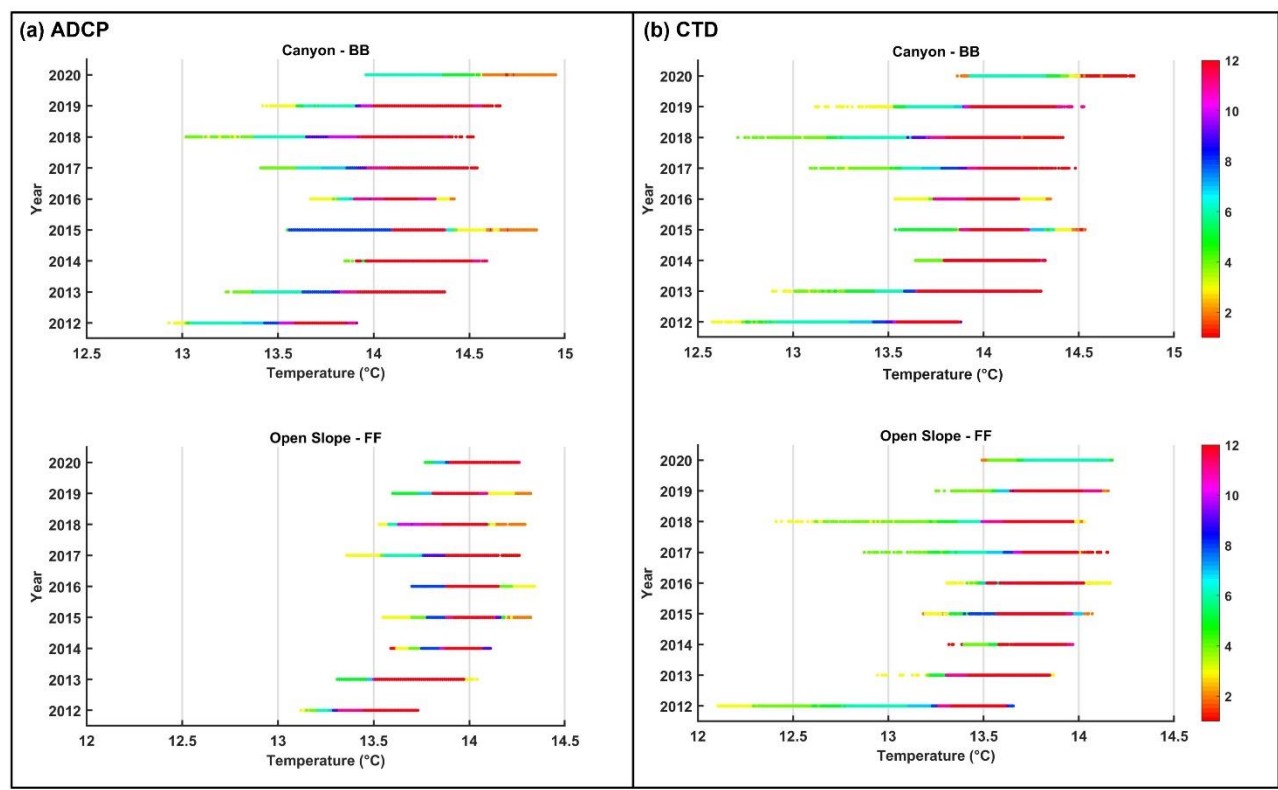

**Figure 6: Scatterplot of ADCP (a) and CTD (b) temperature grouped by years (y-axis) and months (colorscale)**

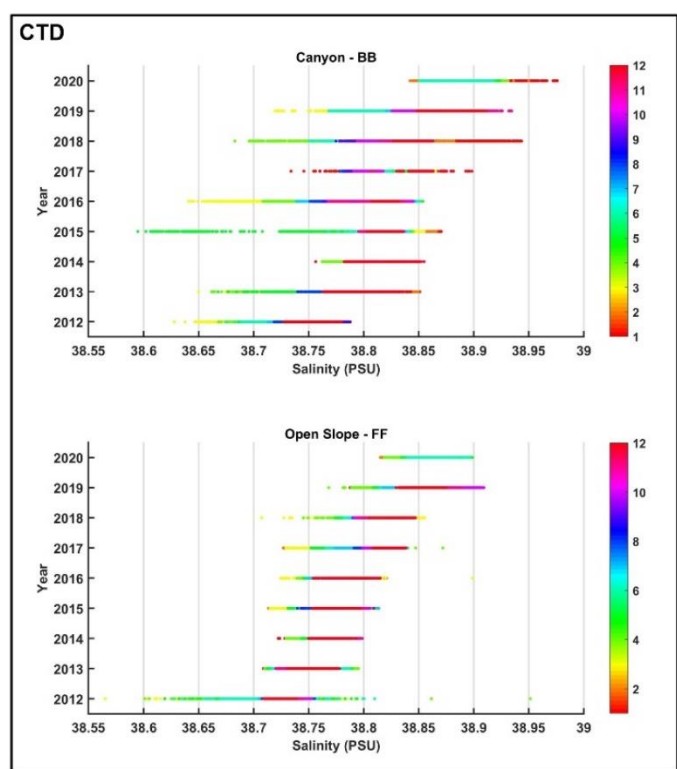

**Figure 7: Scatterplot of salinity records grouped by years (y-axis) and months (colorscale)**


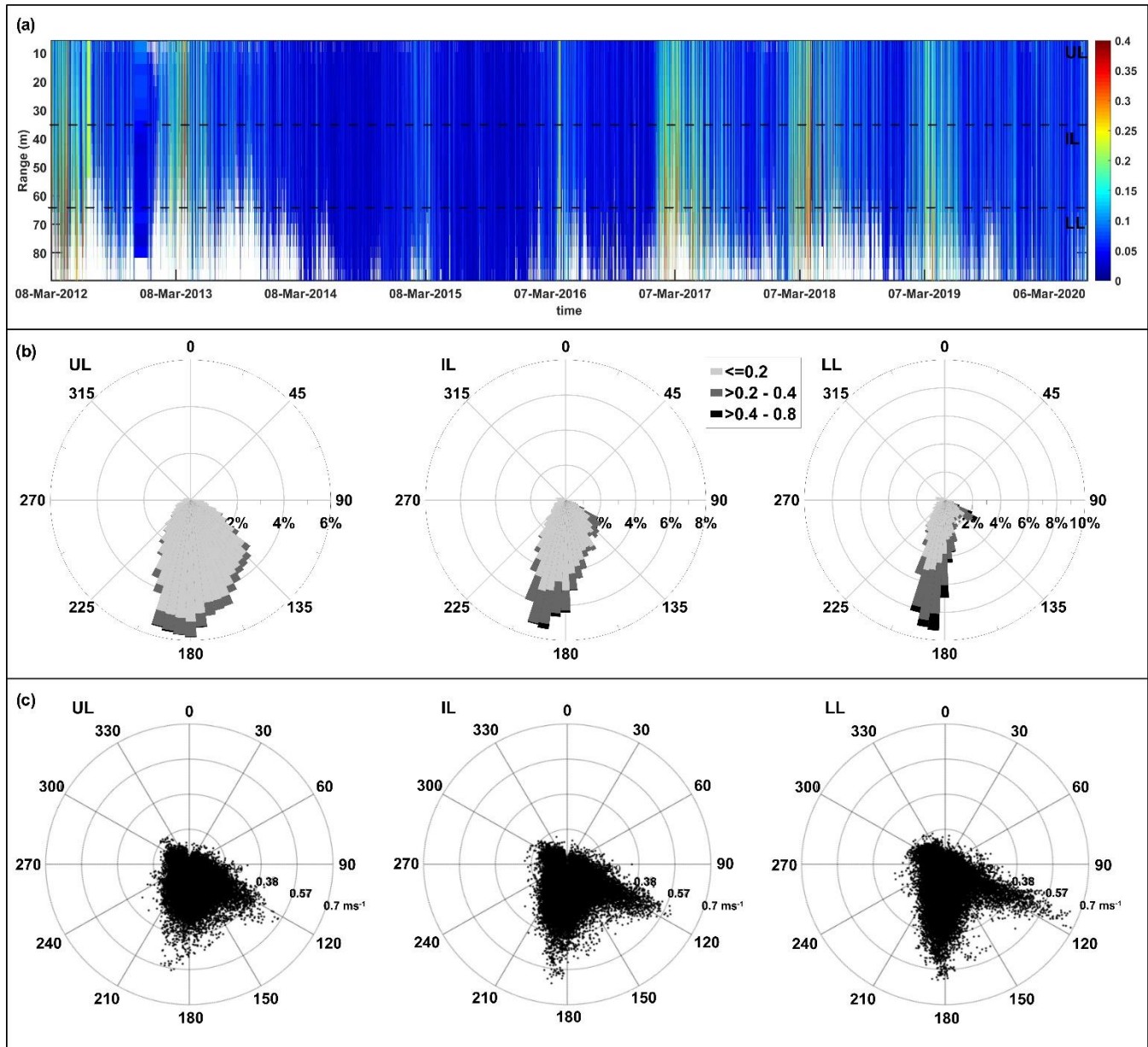

**Figure 8: (a) Currents speed records along the water column (data represented are filtered by 80% percent good), dotted box indicates the water column corresponding to the three layers used for polar plot representation; (b) polar probability plot of current velocity in the three layers of the water column (ms⁻¹); (c) polar scatter plot of observed directional current velocity (ms⁻¹). (UL: Upper Layer; IL: Intermediate Layer; LL: Lower Layer)– BB**


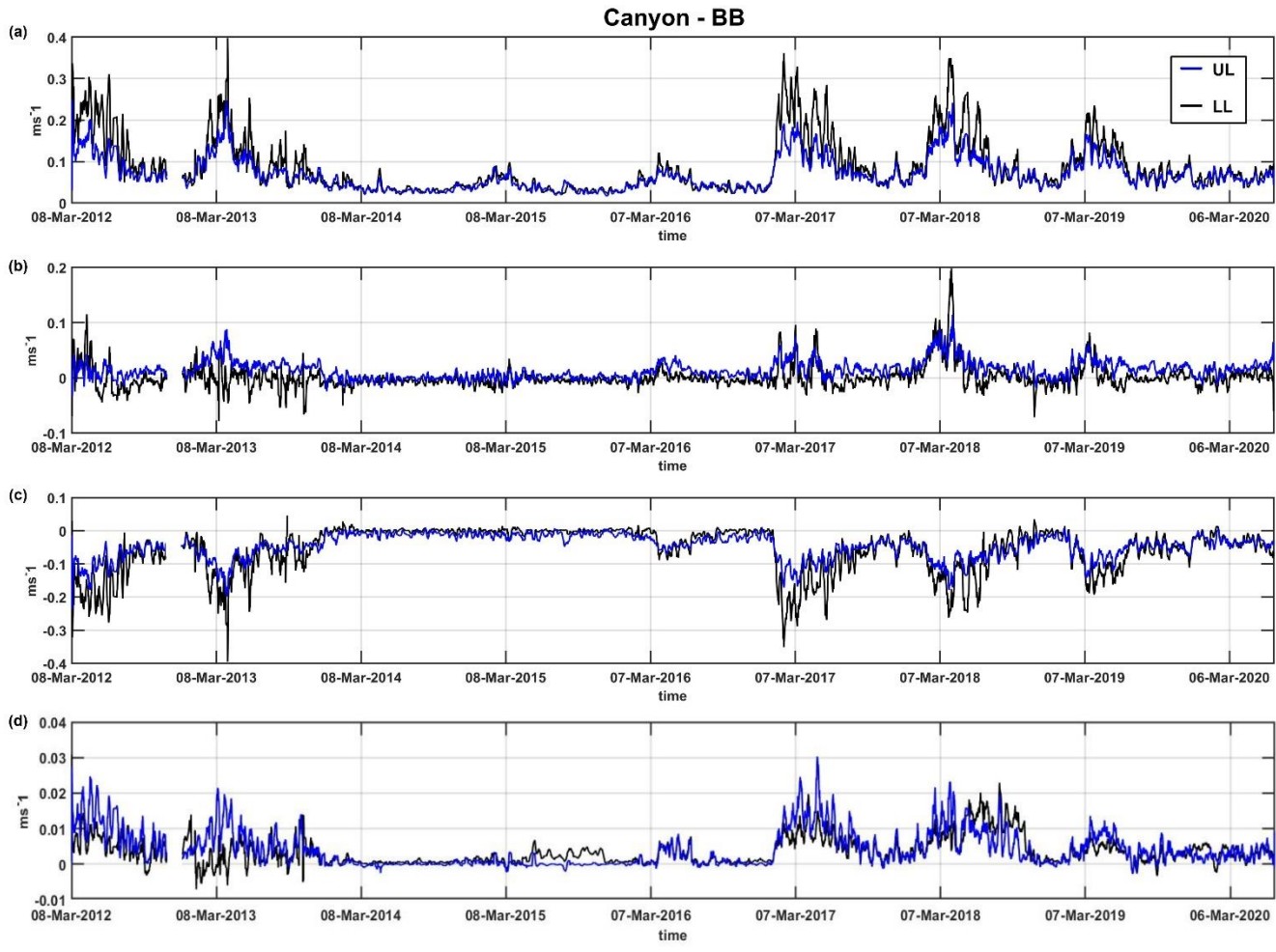


**Figure 9. Time series of currents at BB site in the upper (UL) and lower layer (LL) of the water column: (a) speed, (b) east, (c) north and (d) vertical component – BB. The data are presented with a 7-day smoothing window.**

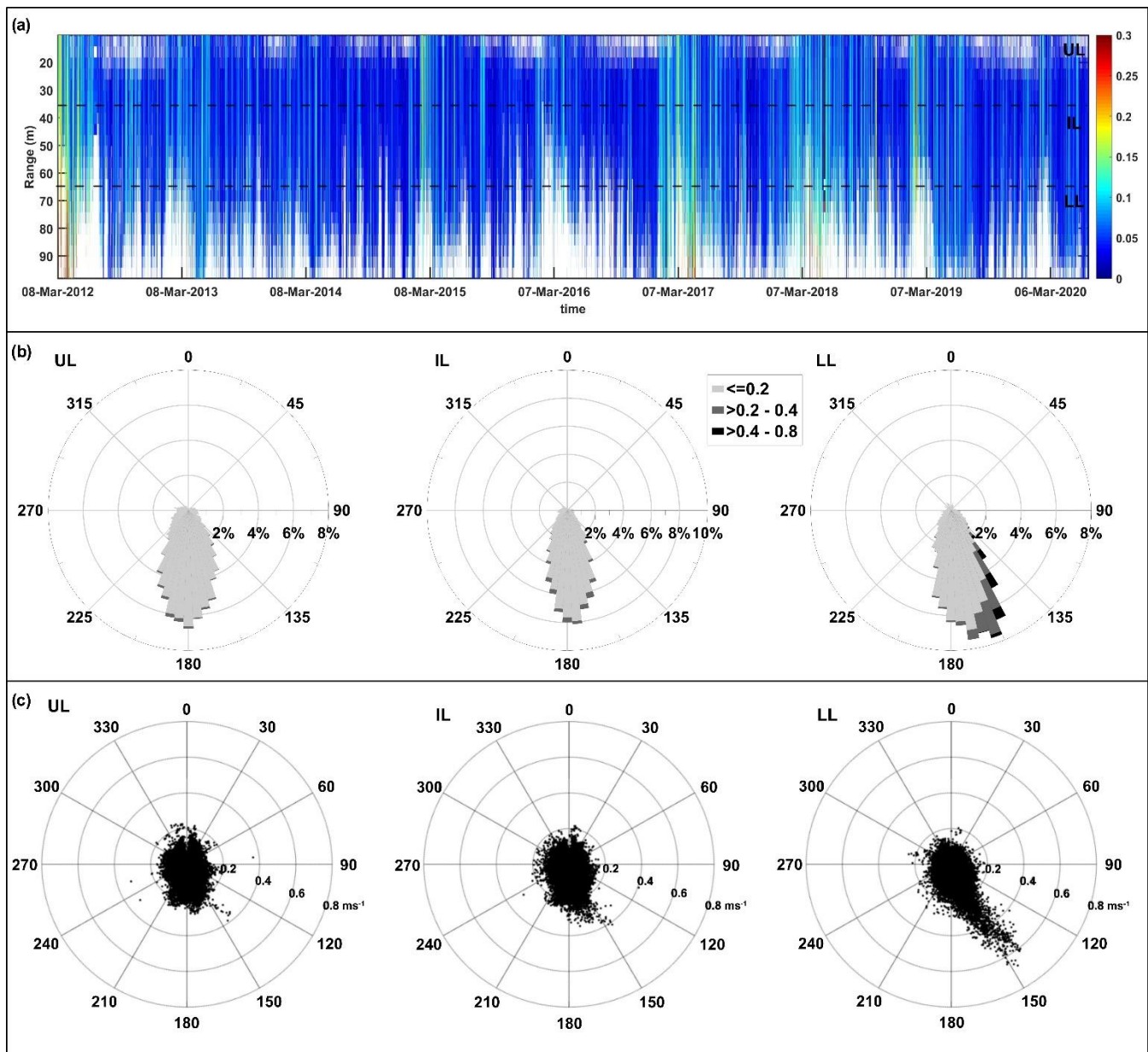

**Figure 10: (a) Currents speed records along the water column (data filtered by 80% percent good), dotted box indicates the water column corresponding to the three layers used for polar plot representation; (b) polar probability plot of current velocity in the three layers of the water column (ms⁻¹); (c) polar scatter plot of observed directional current velocity (ms⁻¹). (UL: Upper Layer; IL: Intermediate Layer; LL: Lower Layer)– - FF.**

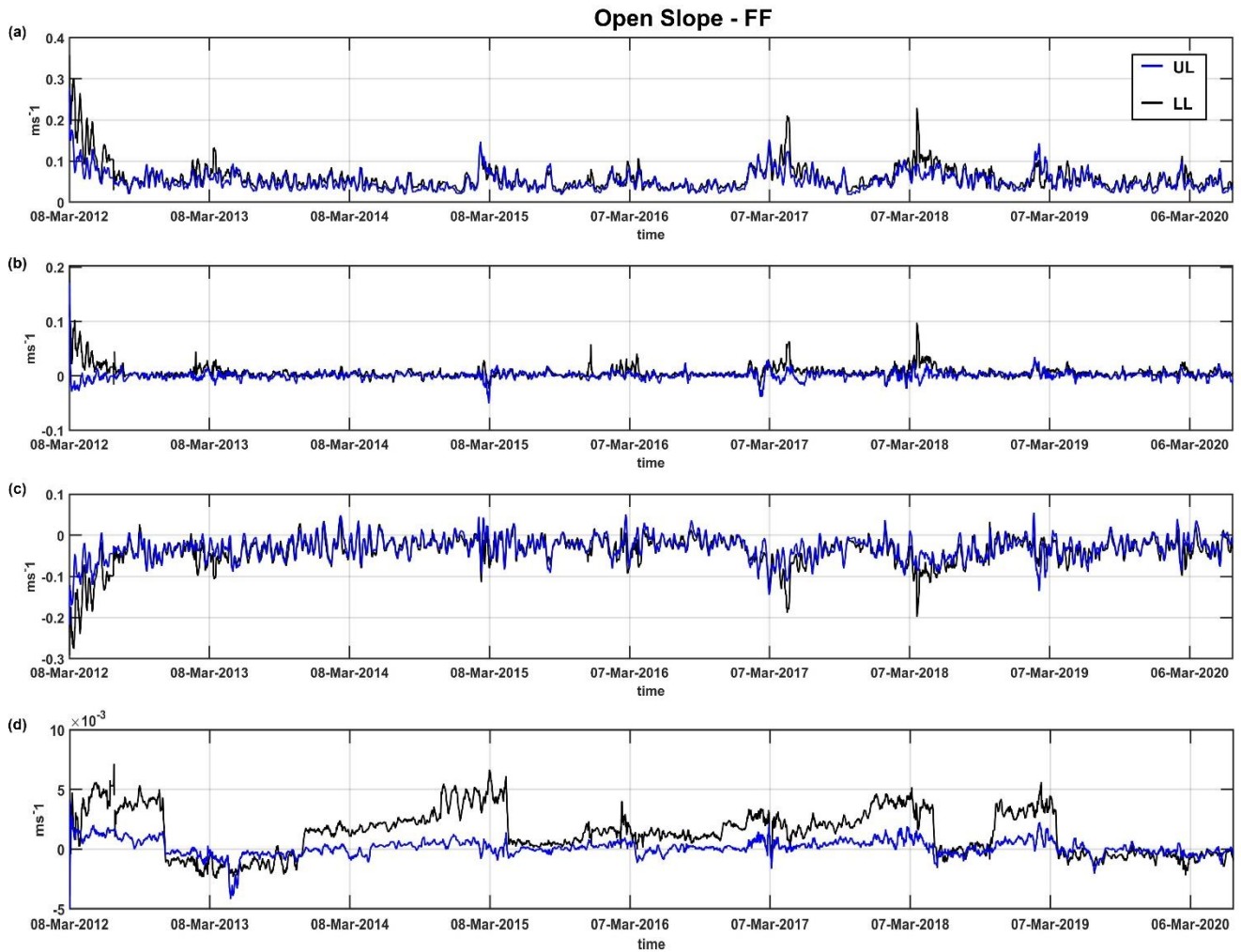

**Figure 11. Time series of currents at FF site in the upper (UL) and lower layer (LL) of the water column: (a) speed, (b) east, (c) north and (d) vertical components -FF. The data are presented with a 7-day smoothing window.**



| N° | Mooring BB | | | | | | Mooring FF | | | | | |
|---|---|---|---|---|---|---|---|---|---|---|---|---|
| | start | end | Mooring position (Lat, Lon) | Depth (m) | ADCP S/N | CTD S/N | start | end | Mooring position (Lat, Lon) | Depth (m) | ADCP S/N | CTD S/N |
| 1 | 08/03/12 | 21/06/12 | 41°20.475'N 17°11.625'E | 504 | 1805 | 6134 | 08/03/12 | 21/06/12 | 41°48.364'N 17°02.292'E | 631 | 6422 | 1709 |
| 2 | 29/06/12 | 06/11/12 | 41°20.478'N 17°11.611'E | 504 | 1805 | 7041 | 30/06/12 | 09/11/12 | 41°48.373'N 17°02.292'E | 631 | 6422 | 7040 |
| 3 | 13/12/12 | 18/04/13 | 41°20.478'N 17°11.605'E | 504 | 6422 | 7040 | 09/11/12 | 14/04/13 | 41°48.367'N 17°02.296'E | 632 | 1805 | 7041 |
| 4 | 18/04/13 | 08/11/13 | 41°20.481'N 17°11.623'E | 504 | 1465 | 7041 | 15/04/13 | 08/11/13 | 41°48.360'N 17°02.292'E | 632 | 1805 | 7266 |
| 5 | 09/11/13 | 09/03/14 | 41°20.471'N 17°11.604'E | 504 | 17316 | 7041 | 08/11/13 | 09/03/14 | 41°48.390'N 17°02.273'E | 632 | 17315 | 7266 |
| 6 | 09/03/14 | 01/11/14 | 41°20.471'N 17°11.628'E | 504 | 17316 | 7040 | 10/03/14 | 02/11/14 | 41°48.390'N 17°02.284'E | 632 | 17315 | 7266 |
| 7 | 01/11/14 | 20/04/15 | 41°20.474'N 17°11.622'E | 500 | 17316 | 7040 | 02/11/14 | 22/04/15 | 41°48.357'N 17°02.297'E | 632 | 17315 | 7266 |
| 8 | 23/04/15 | 09/11/15 | 41°20.456'N 17°11.639'E | 500 | 17315 | 7266 | 22/04/15 | 06/11/15 | 41°48.396'N 17°02.241'E | 636 | 17316 | 7040 |
| 9 | 09/11/15 | 01/04/16 | 41°20.456'N 17°11.639'E | 503 | 17316 | 7041 | 09/11/15 | 02/04/16 | 41°48.396'N 17°02.217'E | 632 | 17315 | 7266 |
| 10 | 05/04/16 | 23/10/16 | 41°20.456'N 17°11.639'E | 497 | 17316 | 6134 | 04/04/16 | 23/10/16 | 41°48.396'N 17°02.217'E | 632 | 17315 | 7266 |
| 11 | 24/10/16 | 23/04/17 | 41°20.456'N 17°11.639'E | 497 | 17316 | 6134 | 24/10/16 | 23/04/17 | 41°48.396'N 17°02.217'E | 632 | 17315 | 7266 |
| 12 | 25/04/17 | 02/11/17 | 41°20.446'N 17°11.620'E | 497 | 17316 | 7041 | 24/04/17 | 02/11/17 | 41°48.402'N 17°02.180'E | 632 | 17315 | 7266 |
| 13 | 04/11/17 | 09/05/18 | 41°20.455'N 17°11.622'E | 497 | 17316 | 7266 | 03/11/17 | 09/05/18 | 41°48.407'N 17°02.186'E | 632 | 17315 | 7041 |
| 14 | 14/05/18 | 07/10/18 | 41°20.471'N 17°11.638'E | 496 | 17315 | 6134 | 09/05/18 | 09/10/18 | 41°48.350'N 17°02.291'E | 632 | 17316 | 7266 |

| 15 | 10/10/18 | 25/03/19 | 41°20.498'N 17°11.617'E | 500 | 17316 | 7266 | 09/10/18 | 24/03/19 | 41°48.224'N 17°02.282'E | 632 | 17315 | 6134 |
|---|---|---|---|---|---|---|---|---|---|---|---|---|
| 16 | 25/03/19 | 19/10/19 | 41°20.491'N 17°11.637'E | 498 | 17316 | 7266 | 24/03/19 | 20/10/19 | 41°48.350'N 17°02.292'E | 632 | 6422 | 6134 |
| 17 | 20/10/19 | 25/06/20 | 41°20.518'N 17°11.645'E | 505 | 17315 | 7266 | 20/10/19 | 26/06/20 | 41°48.316'N 17°02.351'E | 632 | 6422 | 6134 |

**Table 1. Survey details on the two mooring sites. Number of survey (N), start and end of each survey, mooring position, depth of mooring site, ADCP and CTD Serial Number (S/N) of instrument used.**


| CTD S/N | Calibration date |
|---|---|
| 6134 | September 2013 – April 2014 |
| 7041 | March 2012 – February 2014 |
| 7040 | March 2012 – November 2016 |
| 7266 | January 2013 |

**Table 2. Calibration date of the CTD probes named with his Serial Number (S/N).**

| Key | Entry Term | Term definition |
|---|---|---|
| 0 | No quality control | No quality control procedure has been applied |
| 1 | good | Good quality data value. Verified as consistent during quality control process |
| 2 | Probably good value | Data value probably consistent but this is unconfirmed |
| 3 | Probably bad value | Data value recognised inconsistent after quality control |

| | | |
|---|---|---|
| 4 | Bad value | An obviously erroneous data value |
| 5 | Changed value | Data value changed after quality control |
| 6 | Value below detection | The level of the measured phenomenon was too small to be quantified by the technique employed to measure it |
| 7 | Value in excess | The level of the measured phenomenon was too large to be quantified by the technique employed to measure it |
| 8 | Interpolated value | This value has been derived by interpolation from other values in the data object |
| 9 | Missing value | The data value is missing |

**Table 3. Code of data qualifiers flags; are reported the relevant codes to this work. The full list is available at the address:**

**http://seadatanet.maris2.nl/v_bodc_vocab_v2/browse.asp?order=conceptid&formname=search&screen=0&lib=l20**

| MOORING BB | | | | | | | | | | |
|---|---|---|---|---|---|---|---|---|---|---|
| **Month** | **Mean (°C)** | | **SD** | | **Min (°C)** | | **Max (°C)** | | **Δ (°C)** | **Δmax (°C)** |
| | **ADCP** | **CTD** | **ADCP** | **CTD** | **ADCP** | **CTD** | **ADCP** | **CTD** | | |
| **1** | 14.20 | 14.07 | 0.20 | 0.17 | 13.69 | 13.35 | 14.89 | 14.79 | 0.13 | 0.60 |
| **2** | 14.15 | 14.02 | 0.22 | 0.18 | 13.38 | 13.17 | 14.95 | 14.62 | 0.13 | 0.73 |
| **3** | 13.95 | 13.83 | 0.34 | 0.33 | 12.93 | 12.58 | 14.65 | 14.50 | 0.11 | 0.53 |
| **4** | 13.88 | 13.79 | 0.34 | 0.33 | 13.02 | 12.71 | 14.51 | 14.44 | 0.10 | 0.66 |
| **5** | 13.89 | 13.8 | 0.31 | 0.28 | 13.03 | 12.75 | 14.55 | 14.4 | 0.09 | 0.61 |
| **6** | 13.93 | 13.86 | 0.28 | 0.26 | 13.05 | 12.87 | 14.41 | 14.37 | 0.07 | 0.34 |
| **7** | 13.95 | 13.89 | 0.24 | 0.21 | 13.32 | 13.3 | 14.37 | 14.31 | 0.06 | 0.29 |
| **8** | 13.98 | 13.90 | 0.20 | 0.17 | 13.43 | 13.42 | 14.36 | 14.17 | 0.06 | 0.52 |

| Month | Mean (°C) ADCP | CTD | SD ADCP | CTD | Min (°C) ADCP | CTD | Max (°C) ADCP | CTD | Δ (°C) | Δmax (°C) |
|---|---|---|---|---|---|---|---|---|---|---|
| 9 | 14.06 | 13.95 | 0.20 | 0.14 | 13.51 | 13.49 | 14.41 | 14.25 | 0.08 | 0.37 |
| 10 | 14.06 | 13.97 | 0.20 | 0.16 | 13.51 | 13.53 | 14.52 | 14.26 | 0.06 | 0.46 |
| 11 | 14.16 | 14.04 | 0.14 | 0.12 | 13.69 | 13.65 | 14.59 | 14.53 | 0.12 | 0.49 |
| 12 | 14.13 | 14.03 | 0.16 | 0.14 | 13.59 | 13.55 | 14.54 | 14.48 | 0.11 | 0.41 |

| MOORING FF | | | | | | | | | | |
|---|---|---|---|---|---|---|---|---|---|---|
| Month | Mean (°C) | | SD | | Min (°C) | | Max (°C) | | Δ (°C) | Δmax (°C) |
| | ADCP | CTD | ADCP | CTD | ADCP | CTD | ADCP | CTD | | |
| 1 | 13.97 | 13.76 | 0.14 | 0.14 | 13.50 | 13.25 | 14.29 | 14.15 | 0.19 | 0.63 |
| 2 | 13.95 | 13.71 | 0.16 | 0.16 | 13.51 | 13.18 | 14.49 | 14.15 | 0.21 | 0.93 |
| 3 | 13.86 | 13.53 | 0.23 | 0.23 | 13.12 | 12.11 | 14.34 | 14.17 | 0.22 | 1.23 |
| 4 | 13.85 | 13.50 | 0.20 | 0.20 | 13.15 | 12.29 | 14.22 | 14.08 | 0.22 | 1.19 |
| 5 | 13.82 | 13.53 | 0.23 | 0.23 | 13.22 | 12.61 | 14.18 | 14.18 | 0.18 | 0.77 |
| 6 | 13.88 | 13.60 | 0.20 | 0.20 | 13.21 | 12.76 | 14.15 | 14.15 | 0.18 | 0.54 |
| 7 | 13.86 | 13.66 | 0.20 | 0.20 | 13.27 | 13.11 | 14.17 | 14.02 | 0.16 | 0.51 |
| 8 | 13.89 | 13.70 | 0.17 | 0.17 | 13.41 | 13.23 | 14.18 | 14.01 | 0.16 | 0.57 |
| 9 | 13.91 | 13.72 | 0.18 | 0.18 | 13.29 | 13.26 | 14.17 | 14.05 | 0.17 | 0.44 |
| 10 | 13.92 | 13.74 | 0.17 | 0.17 | 13.36 | 13.27 | 14.21 | 14.05 | 0.17 | 0.51 |
| 11 | 13.92 | 13.74 | 0.17 | 0.17 | 13.32 | 13.26 | 14.26 | 14.11 | 0.16 | 0.49 |
| 12 | 13.93 | 13.75 | 0.16 | 0.16 | 13.45 | 13.33 | 14.25 | 14.03 | 0.16 | 0.54 |

**Table 3. Statistical parameters of temperature records of the two moorings grouped by months. SD indicates Standard Deviation and Δ is the mean difference between temperature measured by ADCP and CTD and Δ$_{max}$ il the maximum difference. The value reported in the table refer to original data (not smoothed)**

| MOORING BB | | | | | | | | | | |
|---|---|---|---|---|---|---|---|---|---|---|
| Month | Mean (°C) | | SD | | Min (°C) | | Max (°C) | | Δ (°C) | Δmax (°C) |
| | ADCP | CTD | ADCP | CTD | ADCP | CTD | ADCP | CTD | | |
| 2012 | 13.47 | 13.46 | 0.19 | 0.22 | 12.93 | 12.58 | 13.91 | 13.88 | 0.02 | 0.45 |
| 2013 | 13.87 | 13.79 | 0.17 | 0.18 | 13.23 | 13.89 | 14.37 | 14.30 | 0.07 | 0.61 |
| 2014 | 14.16 | 13.99 | 0.11 | 0.09 | 13.85 | 13.64 | 14.59 | 14.32 | 0.17 | 0.56 |
| 2015 | 14.25 | 14.10 | 0.17 | 0.12 | 13.55 | 13.54 | 14.85 | 14.53 | 0.16 | 0.57 |
| 2016 | 14.11 | 14.04 | 0.12 | 0.11 | 13.67 | 13.54 | 14.42 | 14.35 | 0.08 | 0.46 |
| 2017 | 13.98 | 13.92 | 0.16 | 0.16 | 13.41 | 13.09 | 14.54 | 14.48 | 0.07 | 0.50 |

| | Mean (°C) | | SD | | Min (°C) | | Max (°C) | | Δ (°C) | Δmax (°C) |
|---|---|---|---|---|---|---|---|---|---|---|
| 2018 | 13.91 | 13.83 | 0.21 | 0.22 | 13.02 | 12.71 | 14.52 | 14.42 | 0.08 | 0.66 |
| 2019 | 14.10 | 14.01 | 0.16 | 0.16 | 13.42 | 13.12 | 14.66 | 14.53 | 0.09 | 0.56 |
| 2020 | 14.33 | 14.22 | 0.15 | 0.12 | 13.96 | 13.86 | 14.95 | 14.79 | 0.11 | 0.73 |

| MOORING FF | | | | | | | | | | |
|---|---|---|---|---|---|---|---|---|---|---|
| **Month** | **Mean (°C)** | | **SD** | | **Min (°C)** | | **Max (°C)** | | **Δ (°C)** | **Δmax (°C)** |
| | ADCP | CTD | ADCP | CTD | ADCP | CTD | ADCP | CTD | | |
| **2012** | 13.48 | 13.26 | 0.11 | 0.19 | 13.12 | 12.11 | 13.73 | 13.66 | 0.16 | 1.23 |
| **2013** | 13.68 | 13.48 | 0.12 | 0.12 | 13.31 | 12.94 | 14.04 | 13.87 | 0.17 | 0.64 |
| **2014** | 13.93 | 13.67 | 0.07 | 0.11 | 13.59 | 13.32 | 14.11 | 13.97 | 0.23 | 0.63 |
| **2015** | 14.01 | 13.79 | 0.08 | 0.10 | 13.55 | 13.18 | 14.32 | 14.07 | 0.24 | 0.93 |
| **2016** | 14.04 | 13.82 | 0.08 | 0.09 | 13.70 | 13.31 | 14.34 | 14.17 | 0.22 | 1.01 |
| **2017** | 13.91 | 13.74 | 0.10 | 0.13 | 13.36 | 12.88 | 14.26 | 14.15 | 0.17 | 1.05 |
| **2018** | 13.90 | 13.69 | 0.10 | 0.15 | 13.53 | 12.41 | 14.29 | 14.03 | 0.21 | 1.23 |
| **2019** | 14.96 | 13.82 | 0.08 | 0.12 | 13.60 | 13.25 | 14.32 | 14.15 | 0.14 | 0.69 |
| **2020** | 14.05 | 13.91 | 0.08 | 0.12 | 13.77 | 13.49 | 14.26 | 14.18 | 0.11 | 0.47 |

**Table 4. Statistical parameters of temperature records of the two moorings grouped by years. SD indicates Standard Deviation and Δ is the mean difference between temperature measured by ADCP and CTD and Δ$_{max}$ il the maximum difference. The value reported in the table refer to original data (not smoothed)**




| Month | MOORING BB | | | | MOORING FF | | | |
|---|---|---|---|---|---|---|---|---|
| | Mean | SD | Min | Max | Mean | SD | Min | Max |
| 1 | 38.84 | 0.04 | 38.73 | 38.98 | 38.80 | 0.05 | 38.71 | 38.89 |
| 2 | 38.84 | 0.03 | 38.75 | 38.95 | 38.80 | 0.05 | 38.71 | 38.89 |
| 3 | 38.81 | 0.06 | 38.63 | 38.93 | 38.77 | 0.06 | 38.57 | 38.90 |
| 4 | 38.80 | 0.06 | 38.66 | 38.93 | 38.77 | 0.06 | 38.60 | 38.95 |
| 5 | 38.80 | 0.05 | 38.60 | 38.92 | 38.77 | 0.06 | 38.62 | 38.90 |
| 6 | 38.81 | 0.04 | 38.68 | 38.92 | 38.78 | 0.05 | 38.63 | 38.90 |
| 7 | 38.80 | 0.03 | 38.71 | 38.87 | 38.78 | 0.04 | 38.64 | 38.86 |
| 8 | 38.80 | 0.04 | 38.72 | 38.87 | 38.79 | 0.04 | 38.71 | 38.87 |
| 9 | 38.81 | 0.03 | 38.74 | 38.89 | 38.80 | 0.05 | 38.71 | 38.90 |
| 10 | 38.81 | 0.03 | 38.73 | 38.89 | 38.80 | 0.05 | 38.71 | 38.91 |
| 11 | 38.83 | 0.03 | 38.75 | 38.93 | 38.80 | 0.05 | 38.71 | 38.89 |
| 12 | 38.83 | 004 | 38.73 | 38.91 | 38.80 | 0.05 | 38.71 | 38.89 |
| Year | Mean | SD | Min | Max | Mean | SD | Min | Max |
| 2012 | 38.73 | 0.03 | 38.63 | 38.79 | 38.71 | 0.01 | 38.57 | 38.95 |
| 2013 | 38.79 | 0.02 | 38.65 | 38.85 | 38.73 | 0.02 | 38.71 | 38.80 |
| 2014 | 38.81 | 0.01 | 38.76 | 38.86 | 38.76 | 0.01 | 38.72 | 38.80 |
| 2015 | 38.82 | 0.03 | 38.60 | 38.87 | 38.78 | 0.01 | 38.71 | 38.81 |
| 2016 | 38.80 | 0.03 | 38.64 | 38.85 | 38.79 | 0.01 | 38.72 | 38.90 |
| 2017 | 38.82 | 0.02 | 38.73 | 38.90 | 38.81 | 0.02 | 38.73 | 38.87 |
| 2018 | 38.83 | 0.02 | 38.68 | 38.94 | 38.81 | 0.01 | 38.71 | 38.86 |
| 2019 | 38.85 | 0.03 | 38.72 | 38.93 | 38.85 | 0.03 | 38.64 | 38.91 |
| 2020 | 38.89 | 0.02 | 38.84 | 38.98 | 38.86 | 0.01 | 38.82 | 38.90 |

Table 5. Statistical parameters of salinity records of the two moorings grouped by months and years. SD indicates Standard Deviation and $\Delta$ is the mean difference between temperature measured by ADCP and CTD and $\Delta_{max}$ il the maximum difference. The value reported in the table refer to original data (not smoothed)