# Peer review of "Deep water hydrodynamic observations of two moorings sites on the continental slope of the Southern Adriatic Sea (Mediterranean Sea)"

_Earth System Science Data, 2022_

## Referee Comment (RC1)

**General Comments:**

The paper is presenting an 8-years dataset of temperature, salinity and current meter measurements from an observatory with two moorings along the continental slope of Southern Adriatic Sea for the study of the deep water masses dynamics. The moorings are equipped with ADCP and CTD probes and are part of an experiment set up in Italy after an extremely cold winter at 2012. The experiment funded by a flagship national Italian project and provided so far unique data and knowledge about the hydrodynamics and thermohaline properties of the last 100m of the water column. The instruments used in the experiment are maintained and the collected data are quality controlled. The data files are in an open data format (NetCDF) and are compliant with the FAIR principles but a CTD dataset is not complete and should be corrected. The two moorings are in operation until today and since 2021 are part of the EMSO-ERIC infrastructure. Below is a list of issues to be addressed before the publication of this paper.

**Specific comments:**

1) It is undoubtedly a unique and useful data set to understand the complex hydrodynamics at the area and I would like to read a few more sentences on the advantages of the long-term high resolution monitoring approach as adopted by CIESM and other networks today. In addition, how other projects or activities can benefit from this data set.

2) Although it is mentioned that the two mornings have joined now the EMDO-ERIC infrastructure, I would suggest authors to provide some more information (such as links or references) showing that these data are integrated in EU data systems. This would enhance their FAIRness because through these systems the data are re-usable by many users and applications. Maybe such links are already included in the references but I could not find them.

3) The second section describes the configuration of the observatory and mentions that every 6 months there is a recovery of the instruments for maintenance. I would suggest authors add here few sentences explaining what maintenance includes because not all readers are familiar with field collection practices. For example, does maintenance includes sensors calibration and biases fixing for both CTD and ADCP? For which parameters? Stressing the importance of such maintenance activities would show more clearly that the long term monitoring data are of high quality and accuracy and this is very important when someone is trying to detect variabilities in long-time scales.

4) At the CTD probe description (line 91) it says: "accuracy of ± 0.1% of full-scale range". I would like to read here what does it mean and why this is important for the T,S data accuracy. In the next sentence, isn't the phrase "The available resolution for conductivity is ± 0.0005 S/m, ± 0.005 °C for temperature" a repetition of the previous sentence?

5) Section 2.2 (metadata description), I could not find any metadata report on Dataset Information (DI) and Variables in Dataset (VD). Is by DI and VD is meant the attributes inside the NetCDF files? If so, please clarify and explain in the paper accordingly or better use terms like attributes instead of metadata report.

6) At the same section 2.2 for metadata, it is not only the DOI that make data FAIR. The scope of the journal is to highlight and emphasize the quality, usability, and accessibility of the datasets. Therefore, it would be useful for the readers if authors could expand more the components that make this data set FAIR  for example F:DOI, metadata; A: zenodo, other data portal or tools?; I: open format like NetCDF and std vocabs; R: open and well described data.

7) At the end of section for data, metadata it is mentioned that standardized vocabularies are used. Could the authors include which vocabs they use ?

8) The Data quality check section should be changes to 2.3.

9) In the above section, please mention what tools are used for the quality control. Are these "in-house" made, commercial or other tools? Are these tools open and shareable? This info could also improve the FAIRness of the dataset.

10) Concerning the quality checks, is there any comparison with existing data or climatologies at the area? Do you plan to include such checks in the future releases of time series? Such comparison are basic components of a QC which helps also to find errors at the data due to instrumental biases. It is a key activity to evaluate the quality of the data and I would suggest authors to include such comparisons in future releases.

11) Line 142: Add here the Table 2 reference. There is no reference for Table 2 in the document. Also add a reference for (SeaDataNet, 2010), for example https://www.seadatanet.org/Standards/Data-Quality-Control.

12) Line 156: By checking the data I understand that the bad data (flag=4) are removed from the published at data set at zenodo. I would suggest to keep these values in the published data set so as the QC can be repeated in future (perhaps with other thresholds). In this way you ensure the re-producibility of your data and of your scientific results making thus your data more FAIR.

13) At the start of the Data availability section, why do you use 2 different links ? They end at the same web page.

14) The text fonts at the left axis of mooring sketch at Figure 1 is not very distinctive. If it is feasible to increase the fonts, it would be useful.

**Comments on data files**

15) I could not find filtered variables in the CTD data NetCDF files, only raw data (cond, temp, psal). The included psal_qc, temp_qc are the quality flags and not the filtered variables. The data files should be corrected and reloaded at zenodo.

16) If only good data are kept (flag=1), why the salinity flags at the CTD files as well as the temperature flags at file BB_600_CTD.nc are 1 and 9 ?

**TEXT editing and improvement**

17) Line 18: I think the term "dynamics" is more correct (e.g. "Adriatic deep-water dynamics" instead of "Adriatic deep-water dynamic".

18) Line 19: delete "since 2012".

19) Line 30: Change "figure" to "Fig." as the journal guidelines require (Figure composition).

20) Line 32: change "indicates" to "indicate".
21) Lines 40, 41: merge the two lines.
22) Line 44: I do not find the reference "Gačić et al., 2002". The same for references "Civitarese et al., 2005" at line 45, "Mihanovic et al., 2013" at line 60. "Vilibic and Supic, 2005" at line 265.
23) Line 49: Is there any project link available to be added?
24) Lines 58, 59: merge the two lines.
25) Line 61: change "Carniel et al. 2016" to "Carniel et al., 2016"
26) Line 63: delete the dot before the word but
27) Line 64: change the "broadens" to "broaden" and add a dot at the end of the sentence.
28) Line 65: What is the IFOM? I think authors could add a list of all acronyms used because not all acronyms are given at the paper.
29) Line 65: "provides" instead of "provide"
30) Line 66: I am not a native English speaker but I think "a unique observatory" instead of "unique observatory" would fit better.
31) Line 76: the link does not work.
32) Line 77: change "Figure 1b" to "Fig. 1b".
33) Line 90: delete one of the two dots.
34) Line 146: add a ":" after the parenthesis.
35) Lines 147, 151: add a comma at the end of the equations.
36) Line 160: change "Figure" to "Fig. 2". Same at lines 169, 177, 179, 191, 206 (x3), 208, 209, 227, 231, 233, 241.
37) Line 162: change "et al." to "et al.,". Same at lines 256.
38) Line 265" change "et al," at "et al.,".
39) Line 276: separate the "answerwill".
40) Page 15, Figure 4: change the x-axis of (b) panel from "Year" to "Temperature ($^{o}$C).
41) Line 45, Table 2: the list is not complete. Authors could modify the caption to indicate that these are the relevant codes to this work. Authors could also add a link also of the SeaDataNet L22 QC flag scheme, as L22 has been updated since 2010 the SeaDataNet guidelines were published.

---

## Author Response (AR1)

REVIEWER 1

We thank the reviewer for this useful comment, which helped us to improve the manuscript.

Q1: It is undoubtedly a unique and useful data set to understand the complex hydrodynamics at the area and I would like to read a few more sentences on the advantages of the long-term high resolution monitoring approach as adopted by CIESM and other networks today. In addition, how other projects or activities can benefit from this data set.

A1: We have added a sentence to address this about the advantage of long-term high-resolution monitoring approach and its benefit that this activity can produce to other projects. (Lines 295-299 & 325-331)

Q2: Although it is mentioned that the two mornings have joined now the EMSO-ERIC infrastructure, I would suggest authors to provide some more information (such as links or references) showing that these data are integrated in EU data systems. This would enhance their FAIRness because through these systems the data are re-usable by many users and applications. Maybe such links are already included in the references but I could not find them.

A2: The two sites are part of EMSO-ERIC from 10/2021 but the website of EMSO-ERIC is not yet updated with the BB and FF site information. The text in the manuscript has been changed with more precise information about this. (Lines 65-68)

Q3: The second section describes the configuration of the observatory and mentions that every 6 months there is a recovery of the instruments for maintenance. I would suggest authors add here few sentences explaining what maintenance includes because not all readers are familiar with field collection practices. For example, does maintenance includes sensors calibration and biases fixing for both CTD and ADCP? For which parameters? Stressing the importance of such maintenance activities would show more clearly that the long-term monitoring data are of high quality and accuracy and this is very important when someone is trying to detect variabilities in long-time scales.

A3: The section has been expanded with more detailed information about maintenance, calibration and other instrument procedure. (Lines 103-114); It is also added the table 2 with the date of calibration of each CTD probe. In this part we have mentioned two cruise reports added in the reference list.

Q4: At the CTD probe description (line 91) it says: "accuracy of ± 0.1% of full-scale range". I would like to read here what does it mean and why this is important for the T,S data accuracy. In the next sentence, isn't the phrase "The available resolution for conductivity is ± 0.0005 S/m, ± 0.005 °C for temperature" a repetition of the previous sentence?

A4: The original sentence was confused and in the revised version we wrote a new one about the description of sensors accuracy. (Lines 94-98)

Q5: Section 2.2 (metadata description), I could not find any metadata report on Dataset Information (DI) and Variables in Dataset (VD). Is by DI and VD is meant the attributes inside the NetCDF files? If so, please clarify and explain in the paper accordingly or better use terms like attributes instead of metadata report.

A5: There isn't any metadata report but only the attributes inside the NetCDF files. The metadata description is modified in the revised version (lines 131-136).

Q6: At the same section 2.2 for metadata, it is not only the DOI that make data FAIR. The scope of the journal is to highlight and emphasize the quality, usability, and accessibility of the datasets. Therefore, it would be useful for the readers if authors could expand more the components that make this data set FAIR

for example F:DOI, metadata; A: zenodo, other data portal or tools?; I: open format like NetCDF and std vocabs; R: open and well described data

A6: In the revised version we have expanded the FAIR concept related to our dataset (lines 151-157)

Q7: At the end of section for data, metadata it is mentioned that standardized vocabularies are used. Could the authors include which vocabs they use ?

A7: In the metadata in the global attributes reported the source of the keyword's vocabulary (SeaDataNet parameter discovery vocabulary) and also conventions used (OceanSITES v1.4, SeaDataNet_1.0, COARDS, CF-1.6). In the revised version the description is implemented with this information. (lines 154-157)

Q8 The Data quality check section should be changes to 2.3.

A8: The data quality section number is changed to 2.3

Q9: In the above section, please mention what tools are used for the quality control. Are these "in-house" made, commercial or other tools? Are these tools open and shareable? This info could also improve the FAIRness of the dataset

A9: we have better detailed the quality control used with also example for help the understanding of QC code (lines 171 and 176-185 and 197-198).

Q10: Concerning the quality checks, is there any comparison with existing data or climatologies at the area? Do you plan to include such checks in the future releases of time series? Such comparison are basic components of a QC which helps also to find errors at the data due to instrumental biases. It is a key activity to evaluate the quality of the data and I would suggest authors to include such comparisons in future releases.

A10: Yes, in future releases we plan to include comparison with other existing data and climatology, we also added this in the revised version. (line 328)

Q11: Line 142: Add here the Table 2 reference. There is no reference for Table 2 in the document. Also add a reference for (SeaDataNet, 2010), for example https://www.seadatanet.org/Standards/Data-Quality-Control.

A11: In the caption of Table 3 (ex Table 2) we added the link to the L20 Seadatanet qualifier flag table and we have added the suggested reference. Between lines 177 - 180 the description is expanded.

Q12: Line 156: By checking the data I understand that the bad data (flag=4) are removed from the published at data set at zenodo. I would suggest to keep these values in the published data set so as the QC can be repeated in future (perhaps with other thresholds). In this way you ensure the re-producibility of your data and of your scientific results making thus your data more FAIR

A12: After QC applied by PG80 criterion the value that exceed the threshold are changed to NaN and the flag number in column 11 is 5 as described in the table 3. All data are maintained in the dataset and the original data are reported in the column 12-13-14 with the assigned flag 0 because No quality control procedure has been applied. In the revised version a clearer explanation is provided (Lines 180-182)

Q13: At the start of the Data availability section, why do you use 2 different links ? They end at the same web page

A13: One is the website page of the dataset and the other one is the registered DOI. The sentence in the revised version is changed with only one address (Lines 333-334).

Q14: The text fonts at the left axis of mooring sketch at Figure 1 is not very distinctive. If it is feasible to increase the fonts, it would be useful. Comments on data files

A14: The figure is edited as suggested

Q15: I could not find filtered variables in the CTD data NetCDF files, only raw data (cond, temp, psal). The included psal_qc, temp_qc are the quality flags and not the filtered variables. The data files should be corrected and reloaded at zenodo.

A15: In the original manuscript the description is wrong, the column with header -qc for CTD, correspond only to the flag code of each variable. The description is changed in the revised version. (Lines 145-147)

Q16: If only good data are kept (flag=1), why the salinity flags at the CTD files as well as the temperature flags at file BB_600_CTD.nc are 1 and 9 ? TEXT editing and improvement

A16: not only good data are kept but all data are reported except when the probe is outside the water and not at the correct mooring depth. The flag 9 is assigned when data is missing. We have specified this also in the revised version (Lines 197-198)

Q17: Line 18: I think the term "dynamics" is more correct (e.g. "Adriatic deep-water dynamics" instead of "Adriatic deep-water dynamic".

Q-18-40 text editing and reference editing

A17-40 all text corrections has been edited and also the references

Q41: Line 145, Table 2: the list is not complete. Authors could modify the caption to indicate that these are the relevant codes to this work. Authors could also add a link also of the SeaDataNet L22 QC flag scheme, as L22 has been updated since 2010 the SeaDataNet guidelines were published.

A41: I have added the link of L20 and I have modified the citation on the text. The sentence in the revised version has been modified. (Lines 176-182)

REVIEWER 2

We thank the reviewer for these useful comments, which helped us to improve the manuscript. The revised version of the manuscript will contain the following answers to your comments and suggestions.

Q1: The authors describe the configuration of the observatory and 6-months recovery activities. It could be also useful a brief description of the maintenance methodology that is, undoubtedly, more that change batteries and cleaning the instrumentation. I am quite sure that along 8 years, some sensors and instruments have been calibrated and/or replaced. Some explanation about the calibration processes could be very useful in order to reuse the timeseries, especially when they try to detect decadal to interdecadal signals, as they could do in a near future

A1: The description about the recovery activities and sensor calibration and replacing has been expanded in the revised version. We have added also in table 1 the S/N of CTD sensors used (lines 103-114)

Q2: Regarding to the previous point, the instrument description, lines 81-93 (accuracy, etc) could be easily readable if it is displayed in a table. The authors should evaluate the convenience or not of this suggestion

A2: The original description in the revised version has been changed to a new one more clear (Line 94-98)

Q3: In relation to the dataset at zenodo repository, I find and download the 4 netcdf files, but I am not able to find the reports: "dataset information (DI)" neither "variables in dataset (VD)" described in the paper. It could be my fault, but please check it and add them in case. Mentioned FAIR data principles include more

than giving DOIs and making them accessible (downloadable), but provide the relating information to facilitate the reusing of them. Then, these 2 files are important. I also suppose they include descriptions of the vocabularies standards descriptions which are mentioned in the paper.

A3: There isn't any metadata report but only attributes in the netcdf files, there is a misunderstanding due to a not good writing in the original manuscript. The description of the metadata is modified in the revised version (Lines 126-131and within metadata in the global attributes are specified the source of the keywords_vocabulary (SeaDataNet parameter discovery vocabulary) and also conventions used (OceanSITES v1.4,SeaDataNet_1.0,COARDS,CF-1.6). (Lines 154-156)

Text editing and reference editing issues:

A1-7: All text corrections suggested have been edited and the references are corrected and updated

---

## Editor Decision (ED1)

**Deep water hydrodynamic observations of two moorings sites on the continental slope of the Southern Adriatic Sea (Mediterranean Sea)**

Francesco Paladini de Mendoza, Katrin Schroeder, Leonardo Langone, Jacopo Chiggiato, Mireno Borghini, Patrizia Giordano, Giulio Verazzo, and Stefano Miserocchi

Dear Author and co-authors,

The answers to reviewers have been written quickly and in poor English (i.e. A7, A9, …) as the text added to the final version of the manuscript, which is not is not acceptable yet for publication. Even if the reviewers suggestion is to accept as is, I strongly suggest to revise in depth section 2.3 about Quality Control documenting each QC step (some figure could help) and relative quality flag as required by our journal. The data are visualized with filters that do not show the actual data values that a dataset paper should present (asked before opening the discussion), could you please provide an example of the original time series and of the Quality Control procedure (before and after) for CTD and ADCP?

It follows a list of issues that have not been addressed properly.

Lines 60-69: please check the text

Table 1: please improve both the table (i.e. the lines of the two sites do not correspond, the appearance is terrible) and the caption to be self-explanatory. What is S/N?

Table 2: the calibration dates refer to time periods, i.e. *09/2013 – 04/2014* it means from September 2013 to April 2014? Would it be possible to join table 1 and 2 and provide a complete overview of the mooring sites maintenance activities? Again the caption should be self-explanatory, please improve.

Q4 about accuracy has not been addressed properly. You talk about sensors accuracy but you then write resolution: "*Data of water conductivity was measured by sensor, with a resolution of 0.00005 S/m; the water temperature by means of a thermometer, with resolution of 0.0001 °C; the water pressure by means a pressure strain gauge sensor with an accuracy of 0.002% of full-scale range.*" Could you please include sensor accuracy information?

Line 131: I would rephrase "*The metadata information includes Global….*"

Line 134: please specify or add a reference here about the conventions and keywords vocabulary used.

Lines 134-136: please check the phrase.

Lines 154-157: I suggest to rephrase *"The data and metadata specified in the global attributes use the SeaDataNet parameter discovery vocabulary (https://www.seadatanet.org/Standards/Common-Vocabularies) and the conventions: OceanSITES v1.4, SeaDataNet_1.0, COARDS,CF-155 1.6) sufficiently well described to be readily integrated with other data sources. The metadata accurately describe the data ensuring their reusability in future research and their integration with other compatible data sources."*

Section 2.3 → please revise this section addressing the following issues:

Line 159: I suggest to rephrase: *"First check of ADCP and CTD data is a general screening-view;…"* with "*A first visual check of ADCP and CTD data time series gives a quick idea …* "

Line 161: Please improve this phrase and specify the **out of range criteria** applied: *"This screening phase allows to detect anomalous values which are those **out of range with the rest of the series** and helps to exclude from the time series data when systems are outside the water determining the corrected start and end of the time series."*

Which ranges did you consider?

Line 170: Please improve, *t*his is a suggestion: *"The next data processing consists of the application of applies a data quality control criteria based on the parameter "percentage good" provided from the recording system*

*"A quality flag is assigned to each observation coded following the SeaDataNet Quality Control guidelines (detailed in the (SeaDataNet, 2010) and in particular referred to and the L20 SeaDataNet Measurand Qualifier Flags (last update at address http://seadatanet.maris2.nl/v_bodc_vocab_v2/browse.asp?order=conceptid&formname=search&screen=0&lib=l20) as reported in Table 3.*

Table 3 is not necessary since the link is pointing to the same table, do you want to keep it?

Lines 174-175: *"The data matrix structure explained in the metadata of the published database is composed both by data not subjected to quality control and by data adjusted after quality control."* This phrase is very confused here. At line 182 you explain that the original data are also provided with quality flag0.

Lines 180-185: All the data should be included in the time series and not deleted, quality flag are assigned in order to give the user the quality filter criteria to skip what is not matching the needed quality standards. Moreover, flag 5 is used when data value are adjusted during quality control (i.e. in CMEMS Argo data) and not changed to NaN, thus deleted. This strongly limits the data accessibility and reusability. The practice is to leave the data and assign flag 4 (bad data) if not passing your QC. The question is: what's the difference between the data flagged 1 and the original data flagged 0? The data that passed your PG80 are not modified, right? Please clarify.

What **coarse errors are corrected** by the SBE Data Processing™ software?

Spike test and gradient test did not detect anomalies. This means that the threshold set in your procedure (identical to the SeaDataNet manual, as the added text added) are not proper for your data. Usually a statistical analysis of your dataset on the property (i.e. temperature and salinity gradient) distribution and frequency is necessary to identify the proper thresholds. Could you please provide some stats and justify your choices?

Line 206: you refer to the canyon site, please as I asked at the beginning refer to BB and FF to help the reader. Here your answer to my early question:

Q. please define once canyon site (BB) and open slope site (FF) and keep them in all manuscript.

A. Complete revision is done

Moreover you display smoothed data and refer to mean and extremes computed from original data that are not in Fig. 2a (axis are tighter than the reported extremes), which is misleading. Same comment at line 209. How do you suggest to proceed? I also raised this issue (Q. The data are visualized with filters that do not show the actual data values a dataset paper should present.) before the discussion but nothing has changed.

Line 226: "…vertical temperature gradient is constrained around 0.05°C and 0.2°C …" Vertical gradient is usually reported as °C/m, are you talking about the temperature differences between upper (ADCP) and lower (CTD)? Please clarify and correct eventually also in the successive data interpretation.

Line 262: Could you please improve the description of figure 7? Daily smoothing vs 7-day smoothing window? The caption can be improved as well.

*Figure 7. Time series of currents at BB site in the upper (UL) and lower layer (LL) of the water column: (a) speed, (b) east, (c) north and (d) vertical components. The data  are presented with a 7-day smoothing window.*

Data Availability → please indicate the full link https://doi.org/10.5281/zenodo.6770202

Line 340: *"The dataset presented conclude in 2020 but monitoring activities are still in progress and future data collected by these stations will be added to an updated version of the repository as advancing of the data collection to convey the progress of oceanographic observations to the scientific community."* Could you provide a data update strategy/frequency? i.e. yearly? every 5 years?

Figure and Table captions can be improved to be complete and self-explanatory. Here a couple of suggestions (in section 3.1 BB and FF do not appear, then they are used in 3.2!! Please harmonize):

*Figure 2: ADCP and CTD temperature records at two mooring sites: (a) BB on canyon (600m depth); (b) FF on the open slope (700m). The data are presented with a 3-day smoothing window.*

*Figure 3: ADCP Salinity records on the two mooring sites BB and FF.*

---

## Author Response (AR2)

Deep water hydrodynamic observations of two mooringssites on the continental slope of the Southern Adriatic Sea (Mediterranean Sea)

- Dear Author and co-authors,
  The answers to reviewers have been written quickly and in poor English (i.e. A7, A9, …) as the text added to the final version of the manuscript, which is not is not acceptable yet for publication. Even if the reviewers suggestion is to accept as is, I strongly suggest to revise in depth section 2.3 about Quality Control documenting each QC step (some figure could help) and relative quality flag as required by our journal. The data are visualized with filters that do not show the actual data values that a dataset paper should present (asked before opening the discussion), could you please provide an example of the original time series and of the Quality Control procedure (before and after) for CTD and ADCP?

A: First of all thanks for your comments and suggestions which have improved the quality of the manuscript.

Based on your suggestions, section 2.3 has been thoroughly revised. The quality control for CTD data has been implemented with more appropriate statistics, and figures of the quality control procedure for both ADCP and CTD data have been added. The filter on temperature and salinity data has been removed and the original time series are shown in the figures.

Also based on your comments regarding quality flags assigned to current data and the simultaneous presence of an original and a modified dataset, we decided to reload a new dataset of ADCP data (BB and FF) on the repository in which only the original current data is present side by side with the flag resulting from quality control. In the case where the current datum has a PG<80 we assigned a flag = 3 since as defined by Table 3 it is not definable as wrong datum but more as "Data value recognized inconsistent after quality control." The dataset thus composed was reloaded on zenodo and a new doi was assigned and this was updated in the paper. However within the repository there is still a trace of this change as one can see both version 1.0 which is the previous one and the subsequent update (v 1.1). The doi provided from now on always points to the latest available version of the dataset so it will always remain the same even if new updates to the dataset are made in the future with the addition of the new series from later years but with each update a new version will be formed without deleting the previous ones.

All the changes made are referred to the line number of trackchange manuscript

It follows a list of issues that have not been addressed properly.

- Lines 60-69: please check the text

A: checked and rephrased (line 60-69)

- Table 1: please improve both the table (i.e. the lines of the two sites do not correspond, the appearance is terrible) and the caption to be self-explanatory. What is S/N?

A: The table 1 and his caption was modified.

- Table 2: the calibration dates refer to time periods, i.e. 09/2013 – 04/2014 it means from September 2013 to April 2014? Would it be possible to join table 1 and 2 and provide a complete overview of the mooring sites maintenance activities? Again the caption should be self-explanatory, please improve.

A: I have changed the date of calibration with the name of the month. The caption is modified. We would prefer to leave table 2 separate from table 1 as we have made attempts but it is clearer to us to leave it separate.

- Q4 about accuracy has not been addressed properly. You talk about sensors accuracy but you then write resolution: "Data of water conductivity was measured by sensor, with a resolution of 0.00005 S/m; the water temperature by means of a thermometer, with resolution of 0.0001 °C; the water pressure by means a pressure strain gauge sensor with an accuracy of 0.002% of full-scale range." Could you please include sensor accuracy information?

A: We have added the accuracy information (line 103 and 104)

- Line 131: I would rephrase "The metadata information includes Global…."

A: done (line 138)

- Line 134: please specify or add a reference here about the conventions and keywords vocabulary used.

A: done (line 141-142)

- Lines 134-136: please check the phrase.

A: Rephrased (line 134-135)

- Lines 154-157: I suggest to rephrase "The data and metadata specified in the global attributes use the SeaDataNet parameter discovery vocabulary …..

A: rephrased as suggested (line 157-164)

Section 2.3 à please revise this section addressing the following issues:

- Line 159: I suggest to rephrase: "First check of ADCP and CTD data is a general screening-view;…" with "A first visual check of ADCP and CTD data time series gives a quick idea … "

A: done (line 166)

- Line 161: Please improve this phrase and specify the out of range criteria applied: "This screening phase allows to detect anomalous values which are those out of range with the rest of the series and helps to exclude from the time series data when systems are outside the water determining the corrected start and end of the time series." Which ranges did you consider?

A: This part is implemented with more information about the screening phase (line 170-172). Because the beginning and end of the time series of the various surveys are generally very close we use the range of variability data before and after the individual series for a preliminary assessment of the congruence of the dataset.

- Line 170: Please improve, this is a suggestion: "The next data processing …..

A: done (line 179-182)

- "A quality flag is assigned to each observation ……..

A: rephrased as suggested (line 185-188)

- Table 3 is not necessary since the link is pointing to the same table, do you want to keep it?

A: if is not a problem we prefer to keep it to facilitate the readers.

- Lines 174-175: "The data matrix structure explained in the metadata of the published database is composed both by data not subjected to quality control and by data adjusted after quality control."

This phrase is very confused here. At line 182 you explain that the original data are also provided with quality flag0.

A: The sentence is totally revised and the dataset now contain only the original data (184 – 188)

- Lines 180-185: All the data should be included in the time series and not deleted, quality flag are assigned in order to give the user the quality filter criteria to skip what is not matching the needed quality standards. Moreover, flag 5 is used when data value are adjusted during quality control (i.e. in CMEMS Argo data) and not changed to NaN, thus deleted. This strongly limits the data accessibility and reusability. The practice is to leave the data and assign flag 4 (bad data) if not passing your QC. The question is: what's the difference between the data flagged 1 and the original data flagged 0? The data that passed your PG80 are not modified, right? Please clarify.

A: We have changed the dataset and updated the repository following your suggestion. Only original data are provided together with quality control flag assigned for each observations on the basis of QC procedure (more explanation at line 194-197). No more filtered and original data are provided and no data are deleted from the dataset. In any case the data that passed the QC are not modified.

- What coarse errors are corrected by the SBE Data Processing™ software?

A: This sentence was substantial a generic information about SBE data processing software but not related to the real use of the software with our data. The sentence is changed at line 199 specifing the real use of the software used for data conversion from hexadecimal to ASCII.

- Spike test and gradient test did not detect anomalies. This means that the threshold set in your procedure (identical to the SeaDataNet manual, as the added text added) are not proper for your data. Usually a statistical analysis of your dataset on the property (i.e. temperature and salinity gradient) distribution and frequency is necessary to identify the proper thresholds. Could you please provide some stats and justify your choices?

A: New statistical analysis and QC more suitable to the variability of the data is implemented and added to the text (line 211-218) also with explicative figure. New citation is added in the reference list

- Line 206: you refer to the canyon site, please as I asked at the beginning refer to BB and FF to help the reader. Here your answer to my early question: please define once canyon site (BB) and open slope site (FF) and keep them in all manuscript.

A. I am sorry for failing to properly complete the editing of names in the previous revision but not a complete revision is done

- Moreover you display smoothed data and refer to mean and extremes computed from original data that are not in Fig. 2a (axis are tighter than the reported extremes), which is misleading. Same comment at line 209. How do you suggest to proceed? I also raised this issue (Q. The data are visualized with filters that do not show the actual data values a dataset paper should present.) before the discussion but nothing has changed.

A: the figure 2 and 3 are changed representing the data without filters

- Line 226: "…vertical temperature gradient is constrained around 0.05°C and 0.2°C …" Vertical gradient is usually reported as °C/m, are you talking about the temperature differences between upper (ADCP) and lower (CTD)? Please clarify and correct eventually also in the successive data interpretation.

A: I am talking about the temperature difference. I have changed this in the text (line253-259)

- Line 262: Could you please improve the description of figure 7? Daily smoothing vs 7- day smoothing window? The caption can be improved as well. Figure 7. Time series of currents at BB site in the upper (UL) and lower layer (LL) of the water column: (a) speed, (b) east, (c) north and (d) vertical components. The data for a better visualization are presented with a 7-day smoothing window

In the text I specified that the filter was chosen for better visualization of the data, and the original one is shown in the figure above (Ex. 8a). The image is mainly made to bring out the variability between the layer closer to the bottom and the upper layer. this information was added in the text (line 288-293)

- Data Availability à please indicate the full link https://doi.org/10.5281/zenodo.6770202

A: The new doi is updated and the full link added in the text (line363)

- Line 340: "The dataset presented conclude in 2020 but monitoring activities are still in progress and future data collected by these stations will be added to an updated version of the repository as advancing of the data collection to convey the progress of oceanographic observations to the scientific community." Could you provide a data update strategy/frequency? i.e. yearly? every 5 years?

A: I added a sentence about the intention to update the dataset every 2 years (line 370)

- Figure and Table captions can be improved to be complete and self-explanatory. Here a couple of suggestions (in section 3.1 BB and FF do not appear, then they are used in 3.2!! Please harmonize): Figure 2: ADCP and CTD temperature records at two mooring sites: (a) BB on canyon (600m depth); (b) FF on the open slope (700m). The data are presented with a 3-day smoothing window

A: done

- Figure 3: ADCP Salinity records on the two mooring sites BB and FF

A: done

---

## Author Response (AR3)

Dear Editor,

regarding your comment about the wrong numbering of table I have corrected the numbers of tables as well as I I have updated the correct numbering in the text.

Thanks for your support and review